# Conditioning and Processing: Techniques to Improve Information-Theoretic Generalization Bounds

**Hassan Hafez-Kolahi**
Department of Computer Engineering
Sharif University of Technology
hafez@ce.sharif.edu

**Zeinab Golgooni**
Department of Computer Engineering
Sharif University of Technology
golgooni@ce.sharif.edu

**Shohreh Kasaei**
Department of Computer Engineering
Sharif University of Technology
kasaei@sharif.edu

**Mahdieh Soleymani Baghshah**
Department of Computer Engineering
Sharif University of Technology
soleymani@sharif.edu

## Abstract

Obtaining generalization bounds for learning algorithms is one of the main subjects studied in theoretical machine learning. In recent years, information-theoretic bounds on generalization have gained the attention of researchers. This approach provides an insight into learning algorithms by considering the mutual information between the model and the training set. In this paper, a probabilistic graphical representation of this approach is adopted and two general techniques to improve the bounds are introduced, namely conditioning and processing. In conditioning, a random variable in the graph is considered as given, while in processing a random variable is substituted with one of its children. These techniques can be used to improve the bounds by either sharpening them or increasing their applicability. It is demonstrated that the proposed framework provides a simple and unified way to explain a variety of recent tightening results. New improved bounds derived by utilizing these techniques are also proposed.

## 1   Introduction

Bounding the generalization gap is one of the most studied problems in theoretical machine learning. Classically, uniform bounds on the hypothesis set were one of the most popular techniques in understanding the generalization gap, but they are not always sufficient. In particular that's the case for deep neural networks [22]. Recently there has been a line of research trying to reason about the generalization gap by bounding the mutual information between the dataset and the learned model [17, 16, 18, 6, 3, 19]. Such bounds consider the details of the algorithm used to generate the model according to the dataset, providing new opportunities to take into account the specific dynamics involved in each learning algorithm. In particular, this idea can be used in studying deep neural networks [14, 2, 20]. There is ongoing research to expand this information-theoretic machinery and to tighten the generalization bounds. Bu et al. [7] provided bounds which are based on the mutual information between the model and each individual sample. In another line of work, Asadi et al. [3] used subgaussian assumption to apply chaining techniques in the information-theoretic framework. An idea which was later used to analyze the layered structure of deep neural networks [2]. Recently Steinke and Zakynthinou [19] proposed using a super sample where the training set is selected randomly from it, and computing the mutual information conditioned on this super sample. It was demonstrated how some of the classical tools based on VC-dimension, compression schemes, and differential privacy, can be explained in this new information-theoretic framework. Though in the case of VC theory, the obtained bounds were not tight and have an extra $\log n$ factor, a problem

which authors conjectured could be solved. This work was followed by Haghifam et al. [12], where individual sample bounds and data-dependent bounds were provided in this new setting.

This paper has two main objectives. Firstly, we try to expand the tool set which is used in the information-theoretic bounds by introducing *conditioning* and *processing* techniques. To make the ideas more accessible, we adapt a probabilistic graphical representation of them (made possible by a slight generalization of the standard approach). Using these techniques we explain how bounds similar to previous tightened results can be found. Secondly, we study Chaining Conditional Mutual Information (CCMI) to demonstrate how the conditioning technique can be used to tighten the chaining mutual information of [3], a setting which needs a more elaborate treatment. This brings together the idea of conditioning on a super sample, with the chaining method. We show that this approach eliminates the extra $\log n$ factor which appears in the basic application of the conditioning technique and suggests interesting directions for future work. The proofs of theorems are presented in the supplementary material.

It is worth mentioning that it is also possible to use PAC-Bayesian to derive non-uniform generalization bounds. PAC-Bayesian also has been combined with chaining in [4, 5]. While there are situations where the PAC-Bayesian and information-theoretic approaches become close together (for example see Section 1.3 of [8]), each of these approaches brings their advantages. In particular, in the information-theoretic approach the main focus is on the relation between the learned hypothesis (as a random variable) to other random variables (e.g. dataset), while in PAC-Bayesian the focus is on the relation between two distributions on hypothesis set. An advantage of the former approach is that it is simpler to bring in various random variables in the picture and mold the bounds accordingly. This plays an important role in next sections where graphical models with multiple random variables are studied (in particular see Section E). An advantage of the PAC-Bayesian approach is that it explicitly provides bounds that are independent of the data distribution, while this is not the case for the information-theoretic approach (see Section 3.3).

## 2 Notation and Preliminaries

Capital letters $X$, $Y$, and $Z$ are used for random variables taking values in $\mathcal{X}$, $\mathcal{Y}$ and $\mathcal{Z}$ respectively. The superscripts on distributions and expectations are used to describe conditional distributions, e.g. $P_X^z$ indicates the conditional distribution of $X$ given $Z = z$ and $\mathbb{E}_X^z[f(X,z)]$ indicate the expectation of $f(X,z)$ based on this distribution. The subscripts in expectations indicate the random variables upon which the expectation is acting. In many cases where these random variables are clear from the context, the subscript is omitted to prevent cluttering the notation. The KL divergence of distribution $P_X$ from $Q_X$ is denoted as $\mathrm{KL}(P_X \parallel Q_X) = \int \log \frac{P(X)}{Q(X)} dP$. Mutual information is defined as $I(X;Y) = \mathrm{KL}(P_{XY} \parallel P_X \otimes P_Y)$ where $P_X$ and $P_Y$ are marginal distributions of $P_{XY}$. The conditional mutual information is denoted as $I(X;Y|Z) = \mathbb{E}_Z[I^Z(X;Y)]$ in which for all $z$, $I^z(X;Y)$ is the mutual information on the conditioned distributions $P_{XY}^z$, i.e. $I^z(X;Y) = \mathrm{KL}(P_{XY}^z \parallel P_X^z \otimes P_Y^z)$. Throughout the paper, all logarithms are in natural base and all information-theoretic quantities are in nats.

## 3 Conditioning and Processing

Let us begin by restating a very useful lemma presented by [21].

**Lemma 1** ([21]). *Consider random variables $X$ and $Y$ with joint distribution $P_{XY}$ and a function $g : \mathcal{X} \times \mathcal{Y} \to \mathbb{R}$ such that $g(X,Y)$ is $\sigma$-subgaussian under the distribution $P_{\bar{X}\bar{Y}} = P_X \otimes P_Y$[1], then*

$$\left| \mathbb{E}[g(\bar{X}, \bar{Y})] - \mathbb{E}[g(X,Y)] \right| \leq \sqrt{2\sigma^2 I(X;Y)}. \tag{1}$$

While in Lemma 1 the $\mathbb{E}[g(\bar{X}, \bar{Y})]$ is discussed, the result is also valid for $\mathbb{E}[g(X, \bar{Y})]$ if $\bar{Y}$ is independent of $X$ since $P_{\bar{X}\bar{Y}} = P_{\bar{X}Y}$. Actually this is the case usually encountered in studying the generalization gap. This is summarized in a Probabilistic Graphical Model (PGM) represented in Fig. 1 where we defined $G = g(X,Y)$ and $\bar{G} = g(X, \bar{Y})$. In all the figures, we use the convention that nodes sharing the same letter in their name (with different accents, e.g. $Y$ and $\bar{Y}$), share the same

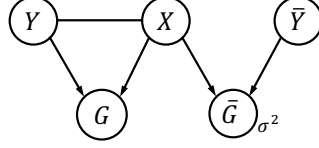

$$\left|\mathbb{E}[\bar{G} - G]\right| \leq \sqrt{2\sigma^2 I(X;Y)}$$

Figure 1: PGM representation for Lemma 2. The direct line between $Y$ and $X$ is used without an arrow to indicate that it can be in either direction. The parameter $\sigma^2$ at the side of node $\bar{G}$ is used to summarize that $\bar{G}$ is $\sigma$-subgaussian.

(conditional) distribution. For example in Fig. 1 we have $P_Y = P_{\bar{Y}}$ and $P_{\bar{G}}^{X\bar{Y}} = P_G^{XY}$. Note that the random variable $G = g(X,Y)$ in Lemma 1 has a deterministic relation with $X$ and $Y$, but the result can be extended to support general stochastic mapping in form of $P_G^{XY}$. This is done in Lemma 2 where we also provided a companion tail bound to set the ground for the next steps.

**Lemma 2.** *Consider random variables $X, Y, \bar{Y}, G$ and $\bar{G}$ with the conditional independences presented in Bayesian network of Fig. 1. Also suppose that $P_{\bar{Y}} = P_Y$ and $P_{\bar{G}}^{X,\bar{Y}} = P_G^{X,Y}$.*

*(a) If the marginal distribution $P_{\bar{G}}$ is $\sigma$-subgaussian, then*

$$\left|\mathbb{E}[\bar{G} - G]\right| \leq \sqrt{2\sigma^2 I(X;Y)}. \tag{2}$$

*(b) If $P_{\bar{G}}^x$ is $\sigma$-subgaussian for all $x \in \mathcal{X}$, then*

$$\Pr\left\{u \times (\bar{G} - G) \geq \epsilon\right\} \leq \frac{4\sigma^2(I(X;Y) + 1)}{\epsilon^2} \quad ; \quad u \in \{-1, 1\}. \tag{3}$$

*Remark:* Note that since in part (b) the bounds for $u = -1$ and $u = 1$ are the same, using union bound we have

$$\Pr\left\{|\bar{G} - G)| \geq \epsilon\right\} \leq \frac{8\sigma^2(I(X;Y) + 1)}{\epsilon^2}. \tag{4}$$

The conditioning and processing are general techniques that can be used to improve information-theoretic bounds such as the ones presented in Lemma 2.

## 3.1 Conditioning

**Lemma 3.** *(Conditioning) Consider random variables $X, Y, Z, H$.*

*(a) Suppose there is a concave function $b : \mathbb{R}_+ \to \mathbb{R}$ which satisfies*

$$\forall z \in \mathcal{Z}; \mathbb{E}^z[H] \leq b(I^z(X;Y)), \tag{5}$$

*then*

$$\mathbb{E}[H] \quad \leq \quad \mathbb{E}_Z[b(I^Z(X;Y))] \tag{6}$$
$$\leq \quad b(I(X;Y|Z)). \tag{7}$$

*(b) Suppose there is a function $\delta : \mathbb{R}_+ \times \mathbb{R} \to \mathbb{R}$ which is concave on its first argument and satisfies*

$$\forall z \in \mathcal{Z}, \epsilon \in \mathbb{R}_+; \Pr^z[H \geq \epsilon] \leq \delta(I^z(X;Y), \epsilon),$$

*then*

$$\Pr[H \geq \epsilon] \quad \leq \quad \delta(I(X;Y|Z), \epsilon). \tag{8}$$

In simple words, Lemma 3 states that if we have a concave information-theoretic bound which is valid in a conditioned setting, there is an accompanying bound on the unconditioned setting which depends on the conditional mutual information. For example by considering $H = \bar{G} - G$, Lemma 3 can be used in conjunction with Lemma 2 in cases that $\bar{Y}$ is not independent of $Y$, but for some random variable $Z$ there is the conditional independence $\bar{Y} \perp\!\!\!\perp Y|Z$ (see Fig. 2a). Though it should be noted that even in the case where $\bar{Y}$ and $Y$ are independent, if $I(X;Y|Z) \leq I(X;Y)$, this lemma can be used to achieve tighter bounds. Conditioning does not necessarily decrease mutual information. The next theorem summarizes some conditions which guarantee that conditioning will not increase the mutual information (see Corollaries of Theorem 2.8.1 in [9]).

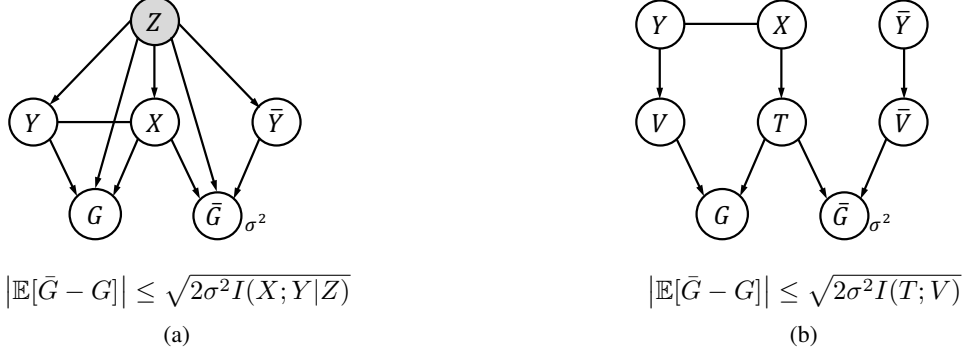

$$\left|\mathbb{E}[\bar{G} - G]\right| \leq \sqrt{2\sigma^2 I(X;Y|Z)}$$

(a)

$$\left|\mathbb{E}[\bar{G} - G]\right| \leq \sqrt{2\sigma^2 I(T;V)}$$

(b)

Figure 2: PGM representation of (a) conditioning and (b) processing. The gray background is used to indicate that a random variable is conditioned.

**Theorem 4** ([9]). *If the random variables $X, Y$ and $Z$ form a Markov chain (in any order), then*

$$I(X;Y|Z) \leq I(X;Y). \tag{9}$$

In the Bayesian network representation, the condition of Theorem 4 is satisfied if the sub-network representing independences of $X$, $Y$ and $Z$ contains no v-structure.

### 3.2 Processing

In many cases there are auxiliary random variables $T$ and $V$ which are generated from $X$ and $Y$ respectively and are the only ingredients needed to find $G$. In such cases the bound can be rewritten to use $T, V$ instead of $X, Y$. This is made precise in the next lemma.

**Lemma 5.** *(Processing) Consider random variables $X, Y, \bar{Y}, G, \bar{G}, V, \bar{V}$ and $T$ with the conditional independences presented in the Bayesian network of Fig. 2b. Also suppose that $P_{\bar{Y}} = P_Y$, $P_V^Y = P_{\bar{V}}^{\bar{Y}}$ and $P_{\bar{G}}^{T,\bar{V}} = P_G^{T,V}$.*

*(a) If the marginal distribution $P_{\bar{G}}$ is $\sigma$-subgaussian, then*

$$\left|\mathbb{E}[\bar{G} - G]\right| \leq \sqrt{2\sigma^2 I(T;V)}. \tag{10}$$

*(b) If $P_{\bar{G}}^t$ is $\sigma$-subgaussian for all $t \in \mathcal{T}$, we have*

$$\Pr\left\{u \times (\bar{G} - G) \geq \epsilon\right\} \leq \frac{4\sigma^2(I(T;V) + 1)}{\epsilon^2} \quad ; \quad u \in \{-1, 1\}. \tag{11}$$

We call this transformation "processing" as it moves away from $(X, Y)$ to $G$ by preprocessing $(T, V)$. This is demonstrated in Fig. 2b. While processing is a simple application of Lemma 2 on $T$ and $V$, this identification as a separate step, makes it simpler to follow the reasoning when the problem gets complicated (we will encounter such problems in section 4.1). Note that processing can not increase the bound because of information processing inequality.

*Remark:* Processing can be combined with conditioning, if the conditions are satisfied in case $Z$ is given, e.g. if $G \perp\!\!\!\perp X, Y | T, V, Z$ then $\left|\mathbb{E}[\bar{G} - G]\right| \leq \sqrt{2\sigma^2 I(T;V|Z)}$.

### 3.3 Applicability of the Bounds

One thing to note about information-theoretic bounds is that even if one can show that a resulting bound is sharper, it does not necessarily mean that it is more applicable. The reason is that the information-theoretic bounds rely on the distribution of data, which is usually unknown. Thus, it should be noted that the real benefit of the proposed techniques is to make it possible to easily transform the bounds, without loosening them, until the situation is ready to have a distribution independent bound or a quantity which is easy to estimate. The problems discussed in next sections demonstrate the applicability of these techniques in practice.

# 4 Some Applications of Conditioning and Processing

Recently there has been a series of papers trying to improve the information-theoretic generalization bounds [6, 7, 19, 12].In Section 4.1 it is demonstrated that it is possible to achieve similar bounds by simple application of conditioning and processing steps. The result is a unified and simple framework to study these bounds, demonstrating the potential of the proposed techniques. In Section 4.2 it is demonstrated how basic bounds based on VC-dimension can be obtained.

## 4.1 A Unified View on Information-Theoretic Generalization Bounds

Consider learning problem where the training set $S = \{(x_j, y_j)\}_{j=1}^n \sim \mu^{\otimes n}$ is given and the learned hypothesis $W \in \mathcal{W}$ is generated by conditional distribution $P_W^S$. We use $S_j$ to denote the $j$th sample of $S$. The loss of hypothesis $w$ on the sample $(x, y)$ is represented by $\ell(w, (x, y))$. Generalization gap of hypothesis $w$ is the difference between the expected loss and the average loss on training set

$$\text{gen}(w) = \mathbb{E}_{XY}[\ell(w, (X, Y))] - \frac{1}{n}\sum_{j=1}^n \ell(w, S_j). \tag{12}$$

One subject of interest is expected generalization gap, $\text{gen}(\mu, P_W^S) \triangleq \mathbb{E}_W[\text{gen}(W)]$. This can be described in various ways each describing a special Bayesian network, providing a variety of opportunities to use the conditioning and processing steps to derive new bounds. These are described in the following series of equations.

$$\text{gen}(\mu, P_W^S) = \mathbb{E}_{S,S',W}\left[\frac{1}{n}\sum_{j=1}^n \ell(W, S_j') - \frac{1}{n}\sum_{j=1}^n \ell(W, S_j)\right] \tag{13}$$

$$= \mathbb{E}_{S,S',W,J}[\ell(W, S_J') - \ell(W, S_J)]. \tag{14}$$

In these equations $S' \sim \mu^{\otimes n}$ is an independent copy of $S$ working as test set. In the second equation an auxiliary random variable $J$ is introduced which is uniformly distributed on $\{1, \ldots, n\}$. These equations can be further expanded by utilizing an alternative approach to generate datasets $S$ and $S'$. Suppose $S^{(1)}$ and $S^{(2)}$ are two sets of $n$ i.i.d. samples from $\mu$ and the binary random variable $U_j$ is uniformly distributed on $\{-1, 1\}$ (Rademacher distribution) and defines whether the $j$th training sample is selected from $S^{(1)}$ or $S^{(2)}$, i.e.

$$S_j = \begin{cases} S_j^{(1)} & U_j = -1 \\ S_j^{(2)} & U_j = 1 \end{cases} \quad ; \quad S_j' = \begin{cases} S_j^{(2)} & U_j = -1 \\ S_j^{(1)} & U_j = 1 \end{cases}. \tag{15}$$

We have

$$\text{gen}(\mu, P_W^S) = \mathbb{E}_{S^{(1)},S^{(2)},U,W}\left[\frac{1}{n}\sum_{j=1}^n \ell(W, S_j') - \frac{1}{n}\sum_{j=1}^n \ell(W, S_j)\right] \tag{16}$$

$$= \mathbb{E}_{S^{(1)},S^{(2)},U,W,J}[\ell(W, S_J') - \ell(W, S_J)] \tag{17}$$

$$= \mathbb{E}_{S^{(1)},S^{(2)},U,W,J}[U_J(\ell(W, S_J^{(1)}) - \ell(W, S_J^{(2)}))]. \tag{18}$$

The required ingredients for the following discussion is that for all $w$ the loss function $L_w = \ell(w, (X, Y))$ is $\sigma_\ell$-subgaussian[2]. Define $L_j = \ell(W, S_j)$ and $L_j' = \ell(W, S_j')$. Also define the averages $R = 1/n \sum_{j=1}^n L_j$ and $R' = 1/n \sum_{j=1}^n L_j'$. Note that $R'$ is $\sigma_\ell^2/n$-subgaussian since $L_j'$s are independent.

In Table 1 it is demonstrated how the previous techniques can be applied to study generalization in practice. To do that the graphical model representing each of the equations (13), (14), (16) and (17) are represented in the first column. In each diagram, the nodes with gray background are conditioned. As usual, the (conditional) distribution of nodes sharing the same base name are the same (e.g. $P_{\bar{U}} = P_U$ and $P_{\bar{R}}^{\bar{S}W} = P_R^{SW}$).

The first row corresponds to the original case. Usage of Lemma 1 by [21] to bound the generalization gap was based on this formulation. The tail bound was studied in [6]. The individual sample bound of the second row is the main subject of paper [7] where they show that this technique could result in much sharper bounds compared to the basic bound and the chaining bound of [3]. Applications on noisy iterative learning algorithms were also proposed. In the third row, a set of indices $\mathcal{J}$ of size $m$ is conditioned, which is one of the inequalities presented in [14]. The fourth row correspond to the setting where super sample is given, which was recently studied by [19] and various applications were considered. The fifth row is individual sample variant of the fourth one, proposed by [12].

In the checklist column, the necessary independence requirements which were validated to find the bounds are listed. The conditioned variables are denoted as $Z$. For fourth and fifth rows were the $Z = (S^{(1)}, S^{(2)})$ are given, the requirement that $S' \perp\!\!\!\perp S|Z$ is no longer satisfied, thus an auxiliary random variable $\bar{S}$ is introduced in a way that guarantees $\bar{S} \perp\!\!\!\perp S'|Z$. This results to bounds for $\mathbb{E}[\bar{R} - R]$ and $\mathbb{E}[\bar{L} - L]$ which is not the same as $\text{gen}(\mu, P_W^S) = \mathbb{E}[R' - R] = \mathbb{E}[L' - L]$ that we were looking for. For tail bounds, there is actually one more difficulty. The quantity $\Pr\{R' - R \geq \epsilon\}$ is not the same as the usual quantity of interest $\Pr\{\text{gen}(w) \geq \epsilon\}$ (i.e. the generalization gap estimated by using a test set of size $n$ is not the same as true generalization gap, even though it is an unbiased estimator). This is as far as we can get by just using the general techniques of processing and conditioning to understand the problem. Fortunately, the usual bounds of interest, differ from these bounds by small constant factors. This is further discussed in the supplementary material (see Theorems E.1 and E.2).

Note that the bounds derived in rows 2 to 5 are improved versions of the first row. The Individual Sample Mutual Information (ISMI) bound of the second row is shown to be always tighter than the basic bound of the first row [7]. Similarly, the conditional mutual information $I(U; W|S^{(1)}, S^{(2)}) + 1)$ present in the fourth row has been shown to be never larger than $I(S; W)$ [12]. The amount of improvement achieved by these techniques can be quite large. Actually, there are cases where the basic bound based on $I(S; W)$ is infinite while the improved bounds are finite [7, 19, 12]. The bound provided in the fifth row is shown to be a further tightened variant of the third row [12].

One of the main applications of information-theoretic generalization bounds is to study Stochastic Gradient Langevin Dynamics (SGLD). In SGLD there is a series of models $W_{(1)}, \ldots, W_{(T)}$. At each step $t$, a training sample $S_{U_{(t)}}$ is sampled, where $U_{(t)} \in \{1, \ldots, n\}$ denotes the random index, and noisy gradient descent is used to find $W_{(t)}$ by the update rule

$$W_{(t)} = W_{(t-1)} - \eta_{(t)}\nabla\ell(W_{(t-1)}, S_{U_{(t)}}) + \sigma_{(t)}\xi,$$

where $\eta_{(t)}$ is step size, $\xi$ is noise (usually a Gaussian random variable) and $\sigma_{(t)}$ controls the strength of noise at time step $t$. This noise addition can control the amount of information stored in $W_{(t)}$ from the sample set. Consequently, information-theoretic generalization bounds can be used to bound the generalization gap. This was the idea first used by [15] to study SGLD generalization based on $I(S; W)$, which was followed by a series of tightening results using the conditional variants of the bounds [7, 14, 12].

## 4.2 VC-dimension

In case of binary classification with zero-one loss $\ell(w, (x, y)) = \mathbb{1}(w(x) \neq y)$, bounds similar to those obtained in VC theory can be retrieved by conditioning on the super sample $S^{(1)}, S^{(2)}$. The set of dichotomies of $\mathcal{W}$ on the set $S$ is defined as $D_{\mathcal{W}}(S) \triangleq \{(w(S_j))_{j=1}^n \mid w \in \mathcal{W}\}$. Now since $S^{(1)}, S^{(2)}$ is given, suppose a fixed ordering on $D_{\mathcal{W}}(S^{(1)} \cup S^{(2)})$ and define $K_W$ as the index of $(W(S_j))_{j=1}^n$ in this ordering. $W$ can be processed to $K_W$, since knowing $K_W$ and the target set $S \subset S^{(1)} \cup S^{(2)}$ is enough to calculate the error. Now note that

$$I^{S^{(1)}, S^{(2)}}(S; K_W) \leq H^{S^{(1)}, S^{(2)}}(K_W) \leq \log(\sup_{S^{(1)}, S^{(2)}} |D_{\mathcal{W}}(S^{(1)} \cup S^{(2)})|) \leq 1 + d_{\text{vc}}(\mathcal{W})\log n,$$

where $d_{\text{vc}}(\mathcal{W})$ is the VC-dimension of $\mathcal{W}$. The last inequality is based on the well known exponential relation between the VC-dimension and the growth function $\Pi_{\mathcal{W}}(2n) = \sup_{S^{(1)}, S^{(2)}} |D(S^{(1)} \cup S^{(2)})|$(see for example Section 2.1.3 of [1]). Finally by noticing that zero-one loss is $1/2$-subgaussian, we have

$$|\mathbb{E}[R' - R]| \leq \sqrt{\frac{2\sigma_\ell^2}{n}I(S; K_W|S^{(1)}, S^{(2)})} \leq \sqrt{\frac{1 + d_{\text{vc}}(\mathcal{W})\log n}{2n}}, \tag{19}$$

Table 1: Summary of Sharpened Information-Theoretic Generalization Bounds.

| Diagram | Checklist | Transformations / Bounds | |
|---|---|---|---|
| | $S' \perp\!\!\!\perp S$ <br> $S' \perp\!\!\!\perp W$ | $\lvert \mathbb{E}[R'-R] \rvert \leq \sqrt{\frac{2\sigma_\ell^2}{n} I(S;W)}$ <br> $\Pr\{R'-R \geq \epsilon\} \leq \frac{4\sigma_\ell^2(I(S;W)+1)}{n\epsilon^2}$ | [21] |
| | $S' \perp\!\!\!\perp S \mid Z$ <br> $S' \perp\!\!\!\perp W \mid Z$ <br> $L \perp\!\!\!\perp S \mid S_J W$ | Condition: $J$ <br> Process: $S \to S_J$ <br><br> $\lvert \mathbb{E}[L'-L] \rvert \leq \frac{1}{n}\sum_{j=1}^{n}\sqrt{2\sigma_\ell^2 I(S_j;W)}$ | [7] |
| | $S' \perp\!\!\!\perp S \mid Z$ <br> $S' \perp\!\!\!\perp W \mid Z$ <br> $R_{\mathcal{J}} \perp\!\!\!\perp S \mid S_{\mathcal{J}} W$ | Condition: $\mathcal{J}$ <br> Process: $S \to S_{\mathcal{J}}$ <br><br> $\lvert \mathbb{E}[R'_{\mathcal{J}}-R_{\mathcal{J}}] \rvert \leq \sqrt{2\frac{\sigma_{\mathcal{J}}^2}{m} I(S_{\mathcal{J}};W)}$ | [14] |
| | $\bar{S} \perp\!\!\!\perp S \mid Z$ <br> $\bar{S} \perp\!\!\!\perp W \mid Z$ | Condition: $S^{(1)}, S^{(2)}$ <br><br> $\lvert \mathbb{E}[\bar{R}-R] \rvert \leq \sqrt{\frac{2\sigma_\ell^2}{n} I(U;W \mid S^{(1)},S^{(2)})}$ <br> $\Pr\{\bar{R}-R \geq \epsilon\} \leq \frac{4\sigma_\ell^2(I(U;W\mid S^{(1)},S^{(2)})+1)}{n\epsilon^2}$ | [19] |
| | $\bar{S} \perp\!\!\!\perp S \mid Z$ <br> $\bar{S} \perp\!\!\!\perp W \mid Z$ <br> $L \perp\!\!\!\perp S \mid S_J W$ | Condition: $S^{(1)}, S^{(2)}, J$ <br><br> Process: $S \to S_J$ <br><br> $\lvert \mathbb{E}[\bar{L}-L] \rvert \leq \sqrt{2\sigma_\ell^2 I(U_J;W \mid S^{(1)},S^{(2)},J)}$ | [12] |

and

$$\Pr\left\{R' - R \geq \epsilon\right\} \leq \frac{4\sigma_\ell^2(I(S; K_W | S^{(1)}, S^{(2)}) + 1)}{n\epsilon^2} \leq \frac{2 + d_{\mathrm{vc}}(\mathcal{W})\log n}{n\epsilon^2}. \tag{20}$$

We know that these bounds are not tight since there is an extra $\log n$ term which is not present in the tightest bound provided in VC theory. In the next section we will see how to overcome this.

A similar result based on conditioning the super sample is presented by [19], but their setting does not allow processing and thus that approach does not work for all learning algorithms. More precisely, the results of [19] show that for each hypothesis class with finite $VC$-dimension, there exists an Empirical Risk Minimizer (ERM) which its generalization can be explained by just conditioning the super sample. This is also demonstrated that there are other ERMs which does not have this property. They also conjectured that the $\log n$ factor can be removed by going beyond ERMs to find a suitable learning algorithm. This is in contrast to the standard results of VC theory which work for all learning algorithms.

## 5   Chaining Conditional Mutual Information

In the previous section, conditioning was used along with the base bound of Lemma 2 to achieve tightened results. But it should be noted that the conditioning technique of Lemma 3 is a general method applicable to any concave information-theoretic bound. In this section, it is demonstrated how this technique can be used in combination with the chaining method of [3] to make tighter bounds. We call this technique Chaining Conditional Mutual Information (CCMI). We first state an alternative formulation of the chaining mutual information.

**Theorem 6.** *Assume that $X_{\mathcal{W}} = \{\sqrt{n}\mathrm{gen}(w)\}_{w \in \mathcal{W}}$ is a separable subgaussian process on the bounded metric space $(\mathcal{W}, d)$ and the learned hypothesis $W$ is a deterministic function of $X_{\mathcal{W}}$. Consider the sequence of functions $(\Pi_k)_{k=k_1(\mathcal{W})}^\infty$ where $k_1(\mathcal{W})$ is the largest integer that satisfies $2^{-(k_1(\mathcal{W})-1)} \geq \mathrm{diam}(\mathcal{W})$, and for all $k \geq k_1$, $\Pi_k : \mathcal{W} \to \mathcal{W}$ is a function satisfying $d(w, \Pi_k(w)) \leq 2^{-k}; \forall w \in \mathcal{W}$. Define $\tilde{W}_k = \Pi_k(W)$, we have*

$$\mathrm{gen}(\mu, P_W^S) \leq \frac{1}{\sqrt{n}}6\sqrt{2} \sum_{k=k_1(\mathcal{W})}^\infty 2^{-k}\sqrt{I(\tilde{W}_k; S)}. \tag{21}$$

Recall that random process $\{X_t\}_{t \in T}$ on metric space $(T, d)$ is called subgaussian if $E[X_t] = 0$ for all $t \in T$ and $X_t - X_s$ is a $d(t, s)$-subgaussian random variable for all $t, s \in T$. Separability is a technical assumption defined in the supplementary material and is assumed in the next discussions. There are two differences between Theorem 6 and the result provided by [3]. First of all, it is restated by focusing on the mappings $\Pi_k$, instead of partitions, a modification which makes next discussions simpler. More importantly, the constraint that $\tilde{W}_k$ should be a function of $\tilde{W}_{k+1}$ (enforced in the original formulation by using an "increasing sequence of partitions") is removed. But to do that, it was required to add the assumption that the learning algorithm is deterministic (and increase the constant by a factor of 2). The latter modification allows us to separately optimize each term of the sum in (21) when trying to find tightest bounds. Let us define the rate-distortion function

$$R(D) \triangleq \inf_{\Pi: \mathcal{W} \to \mathcal{W}} I(\tilde{W}; S); \quad \text{s.t. } d(W, \tilde{W}) \leq D \text{ a.s.}, \tag{22}$$

where $\tilde{W} = \Pi(W)$, we have

$$\mathrm{gen}(\mu, P_W^S) \leq \frac{1}{\sqrt{n}}6\sqrt{2} \sum_{k=k_1(\mathcal{W})}^\infty 2^{-k}\sqrt{R(2^{-k})}. \tag{23}$$

Note that since the algorithms is deterministic, $I(\tilde{W}; S) = I(\tilde{W}; W)$ and the optimization of (22) resembles the setting of rate-distortion theory (see Chapter 10 of [9]).[3]

An integral part of the standard method to study empirical process $\{\mathrm{gen}(w)\}_{w \in \mathcal{W}}$ is based on the usage of a super sample $S^{(1)}, S^{(2)}$ as was done in Section 4.1 (for example see Chapter 7 of [13]).

Consider Eq. (18), we have

$$\text{gen}(\mu, P_W^S) = \mathbb{E}[\frac{1}{n}\sum_{j=1}^{n} U_j(\ell(W, S_j^{(1)}) - \ell(W, S_j^{(2)}))] \qquad (24)$$

Now note that when $S^{(1)}$ and $S^{(2)}$ are given, for each $w$, the randomness is only in the Rademacher random variables $U = (U_k)_{k=1}^{n}$. By using Azuma-Hoeffding inequality, $\{\sqrt{n}\text{gen}(w)\}_{w \in \mathcal{W}}$ is subgaussian with distance function

$$d^{S^{(1)}S^{(2)}}(w, w') \triangleq \left[\frac{1}{n}\sum_{(x,y) \in S^{(1)} \cup S^{(2)}}^{n} (\ell(w, (x,y)) - \ell(w', (x,y)))^2\right]^{1/2}. \qquad (25)$$

Thus, for fixed and given $S^{(1)}$, $S^{(2)}$, the conditions of Theorem 6 are satisfied for $\{\sqrt{n}\text{gen}(w)\}_{w \in \mathcal{W}}$ on space $(\mathcal{W}, d^{S^{(1)}S^{(2)}})$. Now the conditioning lemma, can be used to find the bound on unconditioned case as

$$\text{gen}(\mu, P_W^S) \leq \frac{1}{\sqrt{n}}\mathbb{E}_{S^{(1)}S^{(2)}}[6\sqrt{2}\sum_{k=k_1(\mathcal{W})}^{\infty} 2^{-k}\sqrt{I^{S^{(1)}S^{(2)}}(\tilde{W}_k; S)}]. \qquad (26)$$

Note that here $\tilde{W}_k$s can be generated based on the given super sample by solving the optimization

$$R^{S^{(1)}S^{(2)}}(D) \triangleq \inf_{\Pi: \mathcal{W} \to \mathcal{W}} I^{S^{(1)}S^{(2)}}(\tilde{W}; S); \quad \text{s.t. } d^{S^{(1)}S^{(2)}}(W, \tilde{W}) \leq D \text{ a.s.}. \qquad (27)$$

Dependence on the super sample is the main difficulty of this definition. One way to resolve this is to define the uniform bound of

$$R_U(D) \triangleq \sup_{S^{(1)}S^{(2)}} R^{S^{(1)}S^{(2)}}(D), \qquad (28)$$

which can be used in (23) to control generalization.

Next theorem which is based on [11] shows that $R_U(D)$ can be controlled by $d_{\text{vc}}(\mathcal{W})$.

**Theorem 7.** *Suppose $\mathcal{W}$ has VC-dimension $d_{(vc)}(\mathcal{W})$ and $\ell(w, (x, y)) = \mathbb{1}(w(x) \neq y)$. There is a universal constant $C$ such that the following bound holds on $R_U(D)$ defined in (28)*

$$R_U(D) \leq Cd_{(vc)}(\mathcal{W})\log(\frac{C}{D}). \qquad (29)$$

Using this theorem with (26) we have

$$\text{gen}(\mu, P_W^S) \leq \frac{1}{\sqrt{n}}6\sqrt{2}\sum_{k=k_1(\mathcal{W})}^{\infty} 2^{-k}\sqrt{R_U(2^{-k})} = C'\sqrt{\frac{d_{(vc)}(\mathcal{W})}{n}}, \qquad (30)$$

for a universal constant $C'$. This eliminates the previous extra $\log n$ factor and provides the optimal rate of $\mathcal{O}(n^{-1/2})$.

Note that the real benefit of conditioning and processing techniques is to easily transform the information-theoretic bounds without resorting to applying the uniform bound. As such, it should be emphasized that the real highlight of this section was introduction of rate-distortion function $R(D)$ and its conditioned counterpart $R^{S^{(1)}S^{(2)}}(D)$ (equations (22) and (27)), i.e. before applying the final uniform bound. Also, note that it is rather straightforward to combine CCMI with other ideas from Section 4.1 (e.g. by conditioning on both super sample and random index). This is an advantage of the information-theoretic framework over the PAC-Bayesian approach, as was discussed in Section 1. These subjects provide interesting directions for future study.

## 6 Conclusion

In this paper two techniques to improve information-theoretic generalization bounds were studied. Utilizing a probabilistic graphical representation, it was demonstrated that these techniques provide a simple machinery to reason about the generalization gap and to tighten information-theoretic bounds. Then the conditioning technique was used in conjunction with the method of chaining mutual information as a next step toward understanding the generalization gap in the context of information theory. It was demonstrated how using this technique the bounds based on VC-dimension can be achieved. The developed theory to achieve this result introduced some interesting subjects for further study.

## Broader Impact

As a theoretical work, the direct foreseeable impact is on the academic community. In particular, in the field of machine learning, this work should be viewed along a series of works trying to understand learning algorithms from the lens of information theory. The unified framework introduced in this paper provides an intuitive and convenient way to derive tighter generalization bounds. This can increase the ability of researchers to apply information theory to derive bounds for their learning algorithms and to easily communicate their ideas. These contributions have the potential to boost the ongoing paradigm shift toward an information-theoretic understanding of machine learning. It should also be noted that the studied techniques are not limited to deriving generalization bounds, which is evident from the quite general structure of graphical models and techniques. As such, these kinds of graphical representations along the conditioning and chaining techniques may find their way to be used in other fields beside learning theory as well.

## Acknowledgments and Disclosure of Funding

There are no financial conflicts of interest to disclose.

## Footnotes

[1] Recall that a random variable $V$ is $\sigma$-subgaussian if $\log \mathbb{E}[e^{\lambda(V - \mathbb{E}[V])}] \leq \lambda^2 \sigma^2 / 2$ for all $\lambda \in \mathbb{R}$.

[2]Recall that if a random variable $L$ satisfies $a \le L \le b$, it is $(b - a)/2$-subgaussian.

[3]But note that it differs from the usual setting in which the constraint is on the expected distance and the mappings are stochastic.

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
