[Supplementary Material 1]

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

| *(diagram: $J$, $S$, $W$, $S'$, $S_J$, $S'_J$, $L$, $L'_{\sigma_\ell^2}$)* | $S' \perp\!\!\!\perp S\|Z$ <br> $S' \perp\!\!\!\perp W\|Z$ <br> $L \perp\!\!\!\perp S\|S_J W$ | Condition: $J$ <br> Process: $S \to S_J$ <br><br> $\|\mathbb{E}[L'-L]\| \le \frac{1}{n}\sum_{j=1}^n \sqrt{2\sigma_\ell^2 I(S_j;W)}$ | [7] |
| *(diagram: $\mathcal{J}$, $S$, $W$, $S'$, $S_{\mathcal{J}}$, $S'_{\mathcal{J}}$, $R_{\mathcal{J}}$, $R'_{\mathcal{J}\,\sigma_\ell^2/m}$)* | $S' \perp\!\!\!\perp S\|Z$ <br> $S' \perp\!\!\!\perp W\|Z$ <br> $R_{\mathcal{J}} \perp\!\!\!\perp S\|S_{\mathcal{J}}W$ | Condition: $\mathcal{J}$ <br> Process: $S \to S_{\mathcal{J}}$ <br><br> $\|\mathbb{E}[R'_{\mathcal{J}} - R_{\mathcal{J}}]\| \le \sqrt{2\frac{\sigma_\ell^2}{m}I(S_{\mathcal{J}};W)}$ | [14] |
| *(diagram: $\bar{U}$, $S^{(1)}$, $S^{(2)}$, $U$, $\bar{S}$, $S$, $S'$, $W$, $\bar{R}_{\sigma_\ell^2/n}$, $R$, $R'$)* | $\bar{S} \perp\!\!\!\perp S\|Z$ <br> $\bar{S} \perp\!\!\!\perp W\|Z$ | Condition: $S^{(1)}, S^{(2)}$ <br><br> $\|\mathbb{E}[\bar{R} - R]\| \le \sqrt{\frac{2\sigma_\ell^2}{n}I(U;W\|S^{(1)},S^{(2)})}$ <br> $\Pr\{\bar{R}-R \ge \epsilon\} \le \frac{4\sigma_\ell^2(I(U;W\|S^{(1)},S^{(2)})+1)}{n\epsilon^2}$ | [19] |
| *(diagram: $\bar{U}$, $S^{(1)}$, $S^{(2)}$, $U$, $\bar{S}$, $S$, $S'$, $J$, $\bar{S}_J$, $S_J$, $S'_J$, $W$, $\bar{L}_{\sigma_\ell^2}$, $L$, $L'$)* | $\bar{S} \perp\!\!\!\perp S\|Z$ <br> $\bar{S} \perp\!\!\!\perp W\|Z$ <br> $L \perp\!\!\!\perp S\|S_J W$ | Condition: $S^{(1)}, S^{(2)}, J$ <br><br> Process: $S \to S_J$ <br><br> $\|\mathbb{E}[\bar{L} - L]\| \le \sqrt{2\sigma_\ell^2 I(U_J;W\|S^{(1)},S^{(2)},J)}$ | [12] |

and

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

*Proof.* For Part (a), a slight modification of proof of Lemma 1 in [21] is used. Based on the Donsker–Varadhan variational representation of the relative entropy, for any two distributions $\pi$, $\rho$ on a common measurable space $(\Omega, \mathcal{F})$ we have

$$\mathrm{KL}(\pi \parallel \mu) = \sup_F \left\{ \int_\Omega F d\pi - \log \int_\Omega e^F d\rho \right\} \tag{A.3}$$

where supremum is over all measurable functions $F : \Omega \to \mathbb{R}$, such that $e^F \in L^1(\rho)$. Consider the distribution $\pi = P_{XYG} = P_{XY} \otimes P_G^{XY}$ and let $\mu = P_{X\bar{Y}\bar{G}} = P_X \otimes P_Y \otimes P_G^{XY}$. Note that $\mathbb{E}[G]$ and $\mathbb{E}[\bar{G}]$ are calculated based on $\pi$ and $\mu$, respectively. Define the function $f(x, y, g) = g$. For all $\lambda \in \mathbb{R}$, we have

$$
\begin{aligned}
\mathrm{KL}(\pi \parallel \mu) &\geq \mathbb{E}[\lambda f(X, Y, G)] - \log \mathbb{E}[e^{\lambda f(X, \bar{Y}, \bar{G})}] \\
&= \lambda(\mathbb{E}[G] - \log \mathbb{E}[e^{\lambda \bar{G}}]) \\
&\geq \lambda(\mathbb{E}[G] - \mathbb{E}[\bar{G}]) - \frac{\lambda^2 \sigma^2}{2},
\end{aligned}
\tag{A.4}
$$

where the final inequality is due to the $\sigma$-subgaussian assumption on $P_{\bar{G}}$

$$\log \mathbb{E}[e^{\lambda(G - \mathbb{E}[G])}] \leq \lambda^2 \sigma^2 / 2; \quad \forall \lambda \in \mathbb{R}.$$

The inequality (A.4) gives a nonnegative parabola in $\lambda$. For the inequality to hold for all $\lambda$, the discriminant of this parabola must be nonpositive. This implies that

$$\left| \mathbb{E}[\bar{G}] - \mathbb{E}[G] \right| \leq \sqrt{2\sigma^2 \mathrm{KL}(\pi \parallel \mu)}. \tag{A.5}$$

Now noting that

$$
\begin{aligned}
\mathrm{KL}(\pi \parallel \mu) &= \mathrm{KL}(P_{XY} \otimes P_G^{XY} \parallel P_X \otimes P_Y \otimes P_G^{XY}) \tag{A.6} \\
&= \mathrm{KL}(P_{XY} \parallel P_X \otimes P_Y) \tag{A.7} \\
&= I(X;Y) \tag{A.8}
\end{aligned}
$$

concludes the proof of Part (a).

To prove Part (b), first consider that $u = 1$, for $u = -1$ the proof follows similarly. Let us first restate a lemma from [6].

**Lemma A.1** ([6]). *Let $\pi$ and $\mu$ be distributions on a set $\Omega$ and let $E \subseteq \Omega$. Then,*

$$\pi(E) \leq \frac{KL(\pi \parallel \mu) + 1}{\log(1/\mu(E))}. \tag{A.9}$$

Proving Part (b) requires a stronger technique based on an auxiliary distribution. Let $\pi$ be the distribution presented in Fig. 3, i.e. $\pi = P_{XYG\bar{Y}\bar{G}} = P_{XY} \otimes P_G^{XY} \otimes P_Y \otimes P_G^{XY}$. Consider another distribution $\mu$ which is similar to $\pi$ except that the link between $X$ and $Y$ is severed, i.e. $\mu = P_{X'Y'G'\bar{Y}'\bar{G}'} = P_X \otimes P_Y \otimes P_G^{XY} \otimes P_Y \otimes P_G^{XY}$. Note that again we have $KL(\pi \parallel \mu) = I(X;Y)$.

Let $H' = \bar{G}' - G'$ and $E$ be the event that $H' \geq \epsilon$. To find $\mu(E)$, note that when $X' = x'$ is given, $G'$ and $\bar{G}'$ are independent random variables each having the same distributions as $P_{\bar{G}}^x$. As this distribution is assumed to be $\sigma$-subgaussian, for an arbitrary $x' \in \mathcal{X}$, we have

$$\mathbb{E}^{x'}\left[e^{\lambda(\bar{G}' - G' - (\mathbb{E}^{x'}[\bar{G}' - G']))}\right] = \mathbb{E}^{x'}\left[e^{\lambda(\bar{G}' - \mathbb{E}^{x'}[\bar{G}'])}\right] \mathbb{E}^{x'}\left[e^{\lambda(G' - \mathbb{E}^{x'}[G'])}\right] \leq e^{\lambda^2\sigma^2}; \forall \lambda \in \mathbb{R}. \tag{A.10}$$

This means that the conditional distribution $P_{H'}^{x'}$ is subgaussian with the variance proxy $2\sigma^2$. Also note that $\mathbb{E}^{x'}[\bar{G}' - G'] = 0$. Now, by using the well known fact that the tail of any subgaussian distribution around its mean, is dominated by the Gaussian distribution with the same variance proxy, we have

$$\mathrm{Pr}^{x'}\{H' \geq \epsilon\} \leq e^{-\frac{\epsilon^2}{4\sigma^2}}. \tag{A.11}$$

Since this is valid for all $x'$, by taking expectation from both sides we have

$$\mu(E) = \mathrm{Pr}\{H' \geq \epsilon\} = \mathbb{E}_{X'}[\mathrm{Pr}^{X'}\{H' \geq \epsilon\}] \leq e^{-\frac{\epsilon^2}{4\sigma^2}}. \tag{A.12}$$

Now that a bound on $\mu(E)$ is found, Lemma A.1 can be used to control $\pi(E) = \mathrm{Pr}\{\bar{G} - G \geq \epsilon\}$. We have

$$\pi(E) \leq \frac{KL(\pi \parallel \mu) + 1}{\log(1/\mu(E))} \leq \frac{4\sigma^2(I(X;Y) + 1)}{\epsilon^2}. \tag{A.13}$$

$\square$

# B  Proof of Lemma 3

**Lemma 3.** *(Conditioning) Consider random variables $X, Y, Z, H$.*

*(a) Suppose there is a concave function $b : \mathbb{R}_+ \to \mathbb{R}$ which satisfies*

$$\forall z \in \mathcal{Z}; \mathbb{E}^z[H] \leq b(I^z(X;Y)), \tag{B.1}$$

*then*

$$\begin{aligned} \mathbb{E}[H] &\leq \mathbb{E}_Z[b(I^Z(X;Y))] \tag{B.2} \\ &\leq b(I(X;Y|Z))]. \tag{B.3} \end{aligned}$$

*(b) Suppose there is a function $\delta : \mathbb{R}_+ \times \mathbb{R} \to \mathbb{R}$ which is concave on its first argument and satisfies*

$$\forall z \in \mathcal{Z}, \epsilon \in \mathbb{R}_+; \mathrm{Pr}^z[H \geq \epsilon] \leq \delta(I^z(X;Y), \epsilon),$$

*then*

$$\mathrm{Pr}[H \geq \epsilon] \leq \delta(I(X;Y|Z), \epsilon). \tag{B.4}$$

*Proof.* (a) Note that $E[H] = E_Z[E^Z[H]]$. Since inequality (B.1) holds for all $z \in \mathcal{Z}$, by taking expectation on $Z$ from the random upper bound $b(I^Z(X;Y))$, the inequality (B.2) is achieved. Inequality (B.3) is a consequence of the concavity of $b$, the Jensen's inequality, and the definition of conditional mutual information.

(b) This part follows similarly by noting that for any event $E$ we have $\mathbb{E}_Z[\mathrm{Pr}^Z\{E\}] = \mathrm{Pr}\{E\}$. $\square$

Figure 4: PGM representation of processing technique.

## C    Proof of Theorem 4

**Theorem 4** ([9]). *If the random variables $X, Y$ and $Z$ form a Markov chain (in any order), then*

$$I(X;Y|Z) \leq I(X;Y). \tag{C.1}$$

*Proof.* Suppose the conditional dependences of $X, Y$ and $Z$ satisfies a Markov chain $V_1 - V_2 - V_3$ where $V_i \in \{X, Y, Z\}$ and $(V_i)_{i=1}^3$ is a specific ordering of $(X, Y, Z)$. The Markov property results in $V_1 \perp\!\!\!\perp V_3 | V_2$. If $V_2 = Z$, $I(X;Y|Z) = I(V_1; V_3 | V_2) = 0$ has the smallest possible value and thus (C.1) is satisfied. If $V_2 \neq Z$ either $V_2 = X$ or $V_2 = Y$. Because of the symmetry between $X$ and $Y$, it is enough to consider either of these cases and the other follows similarly. Suppose $V_2 = X$, i.e. $Z \perp\!\!\!\perp Y | X$. By chain rule of mutual information, the quantity $I(Y; (X, Z))$ can be decomposed in two ways and we have

$$I(Y;(X,Z)) = I(Y;Z) + I(Y;X|Z) = I(Y;X) + I(Y;Z|X). \tag{C.2}$$

Since $I(Y;Z|X) = 0$, we have

$$I(X;Y|Z) = I(X;Y) - I(Y;Z). \tag{C.3}$$

Thus, (C.1) is obtained according to the nonnegativity of $I(Y;Z)$.    $\square$

## D    Proof of Lemma 5

**Lemma 5.** *(Processing) Consider random variables $X, Y, \bar{Y}, G, \bar{G}, V, \bar{V}$ and $T$ with the conditional independences presented in the Bayesian network of Fig. 4. Also suppose that $P_{\bar{Y}} = P_Y$, $P_V^Y = P_{\bar{V}}^{\bar{Y}}$ and $P_{\bar{G}}^{T,\bar{V}} = P_G^{T,V}$.*

*(a) If the marginal distribution $P_{\bar{G}}$ is $\sigma$-subgaussian, then*

$$\left| \mathbb{E}[\bar{G} - G] \right| \leq \sqrt{2\sigma^2 I(T;V)}. \tag{D.1}$$

*(b) If $P_{\bar{G}}^t$ is $\sigma$-subgaussian for all $t \in \mathcal{T}$, we have*

$$\Pr\left\{ u \times (\bar{G} - G) \geq \epsilon \right\} \leq \frac{4\sigma^2(I(T;V)+1)}{\epsilon^2} \quad ; \quad u \in \{-1, 1\}. \tag{D.2}$$

*Proof.* Note that $\bar{V} \perp\!\!\!\perp (T, V)$ and marginal distributions $P_V$ and $P_{\bar{V}}$ are the same, because $P_Y = P_{\bar{Y}}$ and $P_V^Y = P_{\bar{V}}^{\bar{Y}}$. The result easilly is obtained as the conditions of Lemma 2 are satisfied for random variables $T, V, \bar{V}, G$ and $\bar{G}$.    $\square$

## E    A Unified View on Information-Theoretic Generalization Bounds

In this section results provided in Section 4.1 are proved and discussed. Recall that we assumed for all $w$ the loss function $\ell(w, X)$ is $\sigma_\ell$-subgaussian. For binary classification with zero-one loss $\sigma_\ell = 1/2$. We also defined $L_j = \ell(W, S_j)$, $L_j' = \ell(W, S_j')$, $R = 1/n \sum_{j=1}^n L_j$ and $R' = 1/n \sum_{j=1}^n L_j'$. Note that $R'$ is $\sigma_\ell^2/n$-subgaussian since $L_j'$s are independent. The object of interest is generalization

Figure 5: Basic generalization.

gap of hypothesis $w$, which is the difference between the expected loss and the average loss on training set

$$\text{gen}(w) = \mathbb{E}_{XY}[\ell(w, (X, Y))] - \frac{1}{n} \sum_{j=1}^{n} \ell(w, S_j). \tag{E.1}$$

We want to have bounds on $\text{gen}(\mu, P_W^S) \triangleq \mathbb{E}[\text{gen}(w)]$ and $\Pr\{\text{gen}(w) \geq \epsilon\}; \epsilon \in \mathbb{R}_+$.

### E.1  Basic Generalization Bound

Considering the conditional independences presented in Fig. 5, using Lemma 2 directly results in

$$\text{gen}(\mu, P_W^S) = |\mathbb{E}[R' - R]| \leq \sqrt{\frac{2\sigma_\ell^2}{n} I(S; W)}, \tag{E.2}$$

and

$$\Pr\{R' - R \geq \epsilon\} \leq \frac{4\sigma_\ell^2 (I(S; W) + 1)}{n\epsilon^2}. \tag{E.3}$$

In Section 4.1, we saw that $E[R' - R]$ in the original formulation is equal to $\text{gen}(\mu, P_W^S)$. But the situation is different for $\Pr\{\text{gen}(W) \geq \epsilon\}$ because now the variance also matters. Next theorem proves that for the large enough $N$ this difference just introduces a constant factor to the bound.

**Theorem E.1.** *If $N \geq \frac{2}{\epsilon^2}$ we have*

$$\Pr\{\text{gen}(W) \geq \epsilon\} \leq 2\Pr\{R' - R \geq \epsilon/2\}. \tag{E.4}$$

*Proof.* The result follows by a standard technique in proving generalization bound that introduces an auxiliary ghost sample set (for example see Appendix A.1 of [1]). Define $e_\mu(W) = \mathbb{E}_{XY}[\ell(W, (X, Y))]$. If $\Pr\{e_\mu(W) - R \geq \epsilon\} = 0$, the desired inequality holds. Assume $\Pr\{e_\mu(W) - R \geq \epsilon\} > 0$, we have

$$
\begin{aligned}
\Pr\{R' - R \geq \frac{\epsilon}{2}\} &\geq \Pr\{R' - R \geq \epsilon/2 \text{ and } e_\mu(W) - R \geq \epsilon\} \\
&= \Pr\{R' - R \geq \frac{\epsilon}{2} \mid e_\mu(W) - R \geq \epsilon\} \Pr\{e_\mu(W) - R \geq \epsilon\} \\
&\geq \Pr\{e_\mu(W) - R' \leq \frac{\epsilon}{2} \mid e_\mu(W) - R \geq \epsilon\} \Pr\{e_\mu(W) - R \geq \epsilon\} \\
&\geq (1 - e^{-\frac{1}{2}\epsilon^2 n}) \Pr\{e_\mu(W) - R \geq \epsilon\}. \tag{E.5}
\end{aligned}
$$

First inequality holds because taking intersection with another event, does not increase probability. The second inequality is obtained since when $e_\mu(W) - R \geq \epsilon$ and $e_\mu(W) - R' \leq \frac{\epsilon}{2}$ it is guaranteed that $R' - R \geq \frac{\epsilon}{2}$. In the final inequality, Hoeffding's bound is used, which is possible because $W$ is independent of $S'$. $\qquad\square$

Using Theorem E.1 we have

$$\Pr\{\text{gen}(W) \geq \epsilon\} \leq \frac{32\sigma_\ell^2 (I(S; W) + 1)}{n\epsilon^2}.$$

Figure 6: Conditioning on sample index.

Figure 7: Conditioning on the super sample.

## E.2 Conditioning on Sample Index

Considering Fig. 6, by conditioning on $J$, and processing $S \to S_J$ we have

$$
\begin{aligned}
\text{gen}(\mu, P_W^S) &= \mathbb{E}[L' - L] \\
&\leq |\mathbb{E}[L' - L]| \\
&\leq \mathbb{E}_J[\sqrt{2\sigma_\ell^2 I(S_J; W)}] \\
&= \frac{1}{n} \sum_{j=1}^{n} \sqrt{2\sigma_\ell^2 I(S_j; W)} \quad\quad\quad (\text{E.6}) \\
&\leq \sqrt{2\sigma_\ell^2 I(S_J; W)}. \quad\quad\quad (\text{E.7})
\end{aligned}
$$

Here it is assumed that $W$ is independent of the order of samples in $S$ and thus $I(S_J; W|J) = I(S_J; W)$. In Section 4.1, we saw that $\mathbb{E}[L'-L]$ is one of the equivalent ways to compute $\text{gen}(\mu, P_W^S)$, and no further corrections is needed.

## E.3 Conditioning on the Super Sample

Considering Fig. 7, by conditioning on $S^{(1)}, S^{(2)}$ we have

$$
|\mathbb{E}[\bar{R} - R]| \leq \sqrt{\frac{2\sigma_\ell^2}{n} I(U; W|S^{(1)}, S^{(2)})}, \quad\quad\quad (\text{E.8})
$$

and

$$
\Pr\left\{\bar{R} - R \geq \epsilon\right\} \leq \frac{4\sigma_\ell^2 (I(U; W|S^{(1)}, S^{(2)}) + 1)}{n\epsilon^2}. \quad\quad\quad (\text{E.9})
$$

Note that here we had to use another branch for random variables $\bar{U}, \bar{S}$ and $\bar{R}$ to have the required conditional independence. Moreover, we assumed that the input distribution $\mu$ is continuous, thus $S^{(1)} \cup S^{(2)}$ is a set of $2n$ distinct random variables. As a result, when $S^{(1)} \cup S^{(2)}$ is given, knowing $S$ is the same as knowing $U$ and $I(S; W|S^{(1)}, S^{(2)}) = I(U; W|S^{(1)}, S^{(2)})$. Also note that the discussion on Section 4.1 demonstrated that the distribution of $R$ and $R'$ is the same as in Section E.1. Next theorem shows the relation between the obtained bounds and generalization error.

**Theorem E.2.** *In the setting represented in Fig. 7, we have*

$$
gen(\mu, P_W^S) = 2\,\mathbb{E}[\bar{R} - R]. \quad\quad\quad (\text{E.10})
$$

*Moreover, if zero-one loss is used and $N \geq \frac{64}{\epsilon^2}$, we have*

$$\Pr\{gen(W) \geq \epsilon\} \leq 4 \times \frac{I(U; W|S^{(1)}, S^{(2)}) + 1}{n(\epsilon/8)^2} \tag{E.11}$$

*Proof.* Define $e_\mu(W) = \mathbb{E}_{XY}[\ell(W, (X, Y))]$ and $e_{\hat{\mu}(W)} = \frac{1}{2n} \sum_{(x_i, y_i) \in S^{(1)}, S^{(2)}} \ell(W, (x_i, y_i))$. Note that $e_{\hat{\mu}(W)} = \frac{1}{2}(R + R')$. For expectation of generalization gap, we have

$$\begin{aligned} \text{gen}(\mu, P_W^S) &= \mathbb{E}[R' - R] \\ &= \mathbb{E}[2(e_{\hat{\mu}(W)} - R) \\ &= 2\mathbb{E}[\bar{R} - R], \end{aligned} \tag{E.12}$$

where final equality is due to the fact that $\bar{R}$ is an unbiased estimator for $e_{\hat{\mu}(W)}$, calculated using $n$ samples from $\hat{\mu}$.

For the tail bound we have

$$\begin{aligned} \Pr\{\text{gen}(W) \geq \epsilon\} &\leq 2\Pr\{R' - R \geq \epsilon/2\} \\ &= 2\Pr\{e_{\hat{\mu}(W)} - R \geq \epsilon/4\} \\ &= 2\Pr\{e_{\hat{\mu}(W)} - \bar{R}) + (\bar{R} - R) \geq \epsilon/4\} \\ &\leq 2\Pr\{e_{\hat{\mu}(W)} - \bar{R} \geq \epsilon/8\} + \Pr\{\bar{R} - R \geq \epsilon/8\} \\ &\leq 2\left(2e^{-2n(\frac{\epsilon}{8})^2} + \Pr\{\bar{R} - R \geq \epsilon/8\}\right) \end{aligned} \tag{E.13}$$

First inequality is based on Theorem E.1. The next equality is based on the algebraic relation $e_{\hat{\mu}(W)} = \frac{1}{2}(R + R')$. In second inequality we used union bound and the mathematical relation $A + B \geq c \Rightarrow (A \geq \frac{c}{2}) \vee (B \geq \frac{c}{2})$. In third inequality, Hoeffding's bound for sampling without replacement is used. Now note that for zero-one loss $\sigma_\ell = 1/2$ and by using (E.9), we have

$$\Pr\left\{\bar{R} - R \geq \epsilon/8\right\} \leq \frac{I(U; W|S^{(1)}, S^{(2)}) + 1}{n(\epsilon/8)^2}. \tag{E.14}$$

On the other hand, when $N \geq \frac{64}{\epsilon^2}$ we also have

$$2e^{-2n(\frac{\epsilon}{8})^2} \leq \frac{1}{n(\epsilon/8)^2}. \tag{E.15}$$

This inequality can be verified for $N = \frac{64}{\epsilon^2}$. Which means that it is also valid for larger $N$, because l.h.s approaches zero with exponential rate while the r.h.s has slower rate of $\frac{1}{n}$.

Finally, we have

$$2e^{-2n(\frac{\epsilon}{8})^2} + \Pr\{\bar{R} - R \geq \epsilon/8\} \leq 2 \times \frac{I(U; W|S^{(1)}, S^{(2)}) + 1}{n(\epsilon/8)^2}, \tag{E.16}$$

which concludes the proof. $\qquad\square$

### E.4 Conditioning on the Super Sample and Index

Considering Fig. 6, by conditioning on $J, S^{(1)}, S^{(2)}$, and processing $S \to S_J$ we have

$$\begin{aligned} |\mathbb{E}[\bar{L} - L]| &\leq \mathbb{E}_J[\sqrt{2\sigma_\ell^2 I^J(U_J; W|S^{(1)}, S^{(2)})}] \tag{E.17} \\ &\leq \sqrt{2\sigma_\ell^2 I(U_J; W|S^{(1)}, S^{(2)}, J)} \tag{E.18} \end{aligned}$$

Note that the same argument which was used in Theorem E.2, can be used for datasets with one sample to show that $\text{gen}(\mu, P_W^S) = 2\mathbb{E}[\bar{L} - L]$.

Figure 8: Conditioning on the super sample and index.

# F  Proof of Theorem 6

**Definition 1** (Separable process)**.** *The random process $\{X_t\}_{t \in T}$ is called separable if there is a countable set $T_0 \subseteq T$ such that $X_t \in \lim_{\substack{s \to t \\ s \in T_0}} X_s$ for all $t \in T$ a.s., where $x \in \lim_{\substack{s \to t \\ s \in T_0}} x_s$ means that there is a sequence $(s_n)$ in $T_0$ such that $s_n \to t$ and $x_{s_n} \to x$.*

**Theorem 6.** *Assume that $X_{\mathcal{W}} = \{\sqrt{n}\,gen(w)\}_{w \in \mathcal{W}}$ is a separable subgaussian process on the bounded metric space $(\mathcal{W}, d)$ and the learned hypothesis $W$ is a deterministic function of $X_{\mathcal{W}}$. Consider the sequence of functions $(\Pi_k)_{k=k_1(\mathcal{W})}^{\infty}$ where $k_1(\mathcal{W})$ is the largest integer that satisfies $2^{-(k_1(\mathcal{W})-1)} \geq diam(\mathcal{W})$, and for all $k \geq k_1$, $\Pi_k : \mathcal{W} \to \mathcal{W}$ is a function satisfying $d(w, \Pi_k(w)) \leq 2^{-k}; \forall w \in \mathcal{W}$. Define $\tilde{W}_k = \Pi_k(W)$, we have*

$$gen(\mu, P_W^S) \leq \frac{1}{\sqrt{n}} 6\sqrt{2} \sum_{k=k_1(\mathcal{W})}^{\infty} 2^{-k} \sqrt{I(\tilde{W}_k; S)}. \tag{F.1}$$

*Proof.* This result is based on Theorem 11 of [3]. Here it is restated using notations used in current paper.

**Theorem F.1.** *Assume that $X_{\mathcal{W}} = \{X_w\}_{w \in \mathcal{W}}$ is a separable subgaussian process on the bounded metric space $(\mathcal{W}, d)$. Consider the sequence of functions $(\Pi_k)_{k=k_1(\mathcal{W})}^{\infty}$ where $k_1(\mathcal{W})$ is the largest integer that satisfies $2^{-(k_1(\mathcal{W})-1)} \geq diam(\mathcal{W})$, and for all $k > k_1$, $\Pi_k : \mathcal{W} \to \mathcal{W}$ is a function satisfying $d(w, \Pi_k(w)) \leq 2^{-k}; \forall w \in \mathcal{W}$. Define $\tilde{W}_k = \Pi_k(W)$ for $k \geq k_1$ and $\tilde{W}_{k_1-1} = w_0$ for an arbitrary $w_0 \in \mathcal{W}$. We have*

$$\mathbb{E}[X_W] \leq 3\sqrt{2} \sum_{k=k_1(\mathcal{W})}^{\infty} 2^{-k} \sqrt{I(\tilde{W}_{k-1}, \tilde{W}_k; X_{\mathcal{W}})}. \tag{F.2}$$

Theorem F.1 is stated for a general random process. In Theorem 6, we used $X_W = \{\sqrt{n}\,gen(w)\}_{w \in \mathcal{W}}$. Moreover, knowing $S$, the values $\{\sqrt{n}\,gen(w)\}_{w \in \mathcal{W}}$ are all deterministically calculated. Thus, by data processing inequality $I(\tilde{W}_{k-1}, \tilde{W}_k; X_{\mathcal{W}}) \leq I(\tilde{W}_{k-1}, \tilde{W}_k; S)$. Now the only remained part is

to remove the dependence in $\tilde{W}_{k-1}$ without resorting to the nested partitioning. We have

$$
\begin{aligned}
\mathbb{E}[X_W] &\leq 3\sqrt{2} \sum_{k=k_1}^{\infty} 2^{-k} \sqrt{I(\tilde{W}_{k-1}, \tilde{W}_k; X_{\mathcal{W}})} \\
&= 3\sqrt{2} \sum_{k=k_1}^{\infty} 2^{-k} \sqrt{I(\tilde{W}_k; X_{\mathcal{W}}) + I(\tilde{W}_{k-1}; X_{\mathcal{W}}|\tilde{W}_k)} \\
&\leq 3\sqrt{2} \sum_{k=k_1}^{\infty} 2^{-k} \sqrt{I(\tilde{W}_k; X_{\mathcal{W}}) + I(\tilde{W}_{k-1}; X_{\mathcal{W}})} &\text{(F.3)} \\
&\leq 3\sqrt{2} (\sum_{k=k_1}^{\infty} 2^{-k} \sqrt{I(\tilde{W}_k; X_{\mathcal{W}})} + \sum_{k=k_1}^{\infty} 2^{-k} \sqrt{I(\tilde{W}_{k-1}; X_{\mathcal{W}})}) &\text{(F.4)} \\
&\leq 3\sqrt{2} (\sum_{k=k_1}^{\infty} 2^{-k} \sqrt{I(\tilde{W}_k; X_{\mathcal{W}})} + \sum_{k=k_1-1}^{\infty} 2^{-(k+1)} \sqrt{I(\tilde{W}_k; X_{\mathcal{W}})}) \\
&\leq 3\sqrt{2} (\sum_{k=k_1}^{\infty} 2^{-k} \sqrt{I(\tilde{W}_k; X_{\mathcal{W}})} + \sum_{k=k_1-1}^{\infty} 2^{-k} \sqrt{I(\tilde{W}_k; X_{\mathcal{W}})}) \\
&\leq 6\sqrt{2} \sum_{k=k_1}^{\infty} 2^{-k} \sqrt{I(\tilde{W}_k; X_{\mathcal{W}})}. &\text{(F.5)}
\end{aligned}
$$

Inequality (F.3) is valid based on Theorem 4 because $\tilde{W}_{k-1} \perp\!\!\!\perp \tilde{W}_k | X_{\mathcal{W}}$ (which is a consequence of the deterministic relation between $W$ and $X_{\mathcal{W}}$). In (F.4) we used the mathematical relation $\sqrt{a+b} \leq \sqrt{a} + \sqrt{b}; \forall a,b \in \mathbb{R}_+$. In final inequality we used $I(w_0; X_{\mathcal{W}}) = 0$, because $w_0$ is a deterministic value. $\qquad\square$

## G    Proof of theorem 7

**Theorem 7.** *Suppose $\mathcal{W}$ has VC-dimension $d_{(vc)}(\mathcal{W})$ and $\ell(w, (x,y)) = \mathbb{1}(w(x) \neq y)$. There is a universal constant $C$ such that the following bound holds on $R_U(D)$ defined in (28)*

$$
R_U(D) \leq C d_{(vc)}(\mathcal{W}) \log(\frac{C}{D}). \tag{G.1}
$$

*Proof.* Note that $I^{S^{(1)}S^{(2)}}(\tilde{W}_k, S) \leq H(\tilde{W}_k)$, where $H(\tilde{W}_k)$ is the entropy of random variable $\tilde{W}_k$ and is bounded by logarithm of number of possible values it takes. A possible mapping for producing $\tilde{W}$ is to use centers of a $D$-covering. Thus,

$$
R^{S^{(1)}S^{(2)}}(D) \leq \log N(\mathcal{W}, d^{S^{(1)}S^{(2)}}, D), \tag{G.2}
$$

where $N(\mathcal{W}, d^{S^{(1)}S^{(2)}}, D)$ is the covering number of $(\mathcal{W}, d)$ at the scale $D$. Recall that $d^{S^{(1)}S^{(2)}}(w, w') = 2 \|w - w'\|_{L^2(\hat{\mu})}$. Here $\hat{\mu}$ is the empirical distribution of the superset and

$$
\|w - w'\|_{L^2(\hat{\mu})} = \left[ \frac{1}{n} \sum_{j=1}^{n} (w(x_j) - w'(x_j))^2 \right]^{1/2}. \tag{G.3}
$$

Now we use the following theorem which is based on [10].

**Theorem G.1** (Dudley [10]). *There is a universal constant $C$ such that*

$$
\sup_{\mu} N(\mathcal{W}, \|.\|_{L^2(\mu)}, D) \leq \left( \frac{C}{D} \right)^{C d_{(vc)}(\mathcal{W})}; \quad \forall D < 1.
$$

Using this theorem with inequality (G.2) completes the proof. $\qquad\square$

[Supplementary Material 2]

# Appendices

## A   Proof of Lemma 2

Figure 3: PGM representation for Lemma 2.

**Lemma 2.** *Consider random variables $X, Y, \bar{Y}, G$ and $\bar{G}$ with the conditional independences presented in the Bayesian network of Fig. 3. Also suppose that $P_{\bar{Y}} = P_Y$ and $P_{\bar{G}}^{X,\bar{Y}} = P_G^{X,Y}$.*

*(a) If the marginal distribution $P_{\bar{G}}$ is $\sigma$-subgaussian, then*

$$\left| \mathbb{E}[\bar{G} - G] \right| \leq \sqrt{2\sigma^2 I(X;Y)}. \tag{A.1}$$

*(b) If $P_{\bar{G}}^x$ is $\sigma$-subgaussian for all $x \in \mathcal{X}$, then*

$$\Pr\left\{ u \times (\bar{G} - G) \geq \epsilon \right\} \leq \frac{4\sigma^2 (I(X;Y) + 1)}{\epsilon^2} \; ; \; u \in \{-1, 1\}, \epsilon \in \mathbb{R}_+. \tag{A.2}$$

*Proof.* For Part (a), a slight modification of proof of Lemma 1 in [21] is used. Based on the Donsker–Varadhan variational representation of the relative entropy, for any two distributions $\pi$, $\rho$ on a common measurable space $(\Omega, \mathcal{F})$ we have

$$\mathrm{KL}(\pi \parallel \mu) = \sup_F \left\{ \int_\Omega F d\pi - \log \int_\Omega e^F d\rho \right\} \tag{A.3}$$

where supremum is over all measurable functions $F : \Omega \to \mathbb{R}$, such that $e^F \in L^1(\rho)$. Consider the distribution $\pi = P_{XYG} = P_{XY} \otimes P_G^{XY}$ and let $\mu = P_{X\bar{Y}\bar{G}} = P_X \otimes P_Y \otimes P_G^{XY}$. Note that $\mathbb{E}[G]$ and $\mathbb{E}[\bar{G}]$ are calculated based on $\pi$ and $\mu$, respectively. Define the function $f(x, y, g) = g$. For all $\lambda \in \mathbb{R}$, we have

$$
\begin{aligned}
\mathrm{KL}(\pi \parallel \mu) &\geq \mathbb{E}[\lambda f(X, Y, G)] - \log \mathbb{E}[e^{\lambda f(X, \bar{Y}, \bar{G})}] \\
&= \lambda (\mathbb{E}[G] - \log \mathbb{E}[e^{\lambda \bar{G}}]) \\
&\geq \lambda (\mathbb{E}[G] - \mathbb{E}[\bar{G}]) - \frac{\lambda^2 \sigma^2}{2},
\end{aligned}
\tag{A.4}
$$

where the final inequality is due to the $\sigma$-subgaussian assumption on $P_{\bar{G}}$

$$\log \mathbb{E}[e^{\lambda(G - \mathbb{E}[G])}] \leq \lambda^2 \sigma^2 / 2; \quad \forall \lambda \in \mathbb{R}.$$

The inequality (A.4) gives a nonnegative parabola in $\lambda$. For the inequality to hold for all $\lambda$, the discriminant of this parabola must be nonpositive. This implies that

$$\left| \mathbb{E}[\bar{G}] - \mathbb{E}[G] \right| \leq \sqrt{2\sigma^2 \mathrm{KL}(\pi \parallel \mu)}. \tag{A.5}$$

Now noting that

$$
\begin{aligned}
\mathrm{KL}(\pi \parallel \mu) &= \mathrm{KL}(P_{XY} \otimes P_G^{XY} \parallel P_X \otimes P_Y \otimes P_G^{XY}) \tag{A.6} \\
&= \mathrm{KL}(P_{XY} \parallel P_X \otimes P_Y) \tag{A.7} \\
&= I(X;Y) \tag{A.8}
\end{aligned}
$$

concludes the proof of Part (a).

To prove Part (b), first consider that $u = 1$, for $u = -1$ the proof follows similarly. Let us first restate a lemma from [6].

**Lemma A.1** ([6]). *Let $\pi$ and $\mu$ be distributions on a set $\Omega$ and let $E \subseteq \Omega$. Then,*

$$\pi(E) \leq \frac{KL(\pi \parallel \mu) + 1}{\log(1/\mu(E))}. \tag{A.9}$$

Proving Part (b) requires a stronger technique based on an auxiliary distribution. Let $\pi$ be the distribution presented in Fig. 3, i.e. $\pi = P_{XYG\bar{Y}\bar{G}} = P_{XY} \otimes P_G^{XY} \otimes P_Y \otimes P_G^{XY}$. Consider another distribution $\mu$ which is similar to $\pi$ except that the link between $X$ and $Y$ is severed, i.e. $\mu = P_{X'Y'G'\bar{Y}'\bar{G}'} = P_X \otimes P_Y \otimes P_G^{XY} \otimes P_Y \otimes P_G^{XY}$. Note that again we have $KL(\pi \parallel \mu) = I(X;Y)$.

Let $H' = \bar{G}' - G'$ and $E$ be the event that $H' \geq \epsilon$. To find $\mu(E)$, note that when $X' = x'$ is given, $G'$ and $\bar{G}'$ are independent random variables each having the same distributions as $P_{\bar{G}}^x$. As this distribution is assumed to be $\sigma$-subgaussian, for an arbitrary $x' \in \mathcal{X}$, we have

$$\mathbb{E}^{x'}\left[e^{\lambda(\bar{G}' - G' - (\mathbb{E}^{x'}[\bar{G}' - G']))}\right] = \mathbb{E}^{x'}\left[e^{\lambda(\bar{G}' - \mathbb{E}^{x'}[\bar{G}'])}\right] \mathbb{E}^{x'}\left[e^{\lambda(G' - \mathbb{E}^{x'}[G'])}\right] \leq e^{\lambda^2 \sigma^2}; \forall \lambda \in \mathbb{R}. \tag{A.10}$$

This means that the conditional distribution $P_{H'}^{x'}$ is subgaussian with the variance proxy $2\sigma^2$. Also note that $\mathbb{E}^{x'}[\bar{G}' - G'] = 0$. Now, by using the well known fact that the tail of any subgaussian distribution around its mean, is dominated by the Gaussian distribution with the same variance proxy, we have

$$\mathrm{Pr}^{x'}\{H' \geq \epsilon\} \leq e^{-\frac{\epsilon^2}{4\sigma^2}}. \tag{A.11}$$

Since this is valid for all $x'$, by taking expectation from both sides we have

$$\mu(E) = \mathrm{Pr}\{H' \geq \epsilon\} = \mathbb{E}_{X'}[\mathrm{Pr}^{X'}\{H' \geq \epsilon\}] \leq e^{-\frac{\epsilon^2}{4\sigma^2}}. \tag{A.12}$$

Now that a bound on $\mu(E)$ is found, Lemma A.1 can be used to control $\pi(E) = \mathrm{Pr}\{\bar{G} - G \geq \epsilon\}$. We have

$$\pi(E) \leq \frac{KL(\pi \parallel \mu) + 1}{\log(1/\mu(E))} \leq \frac{4\sigma^2(I(X;Y) + 1)}{\epsilon^2}. \tag{A.13}$$

$\square$

# B   Proof of Lemma 3

**Lemma 3.** *(Conditioning) Consider random variables $X, Y, Z, H$.*

*(a) Suppose there is a concave function $b : \mathbb{R}_+ \to \mathbb{R}$ which satisfies*

$$\forall z \in \mathcal{Z}; \mathbb{E}^z[H] \leq b(I^z(X;Y)), \tag{B.1}$$

*then*

$$\mathbb{E}[H] \quad \leq \quad \mathbb{E}_Z[b(I^Z(X;Y))] \tag{B.2}$$
$$\leq \quad b(I(X;Y|Z))]. \tag{B.3}$$

*(b) Suppose there is a function $\delta : \mathbb{R}_+ \times \mathbb{R} \to \mathbb{R}$ which is concave on its first argument and satisfies*

$$\forall z \in \mathcal{Z}, \epsilon \in \mathbb{R}_+; \mathrm{Pr}^z[H \geq \epsilon] \leq \delta(I^z(X;Y), \epsilon),$$

*then*

$$\mathrm{Pr}[H \geq \epsilon] \quad \leq \quad \delta(I(X;Y|Z), \epsilon). \tag{B.4}$$

*Proof.* (a) Note that $E[H] = E_Z[E^Z[H]]$. Since inequality (B.1) holds for all $z \in \mathcal{Z}$, by taking expectation on $Z$ from the random upper bound $b(I^Z(X;Y))$, the inequality (B.2) is achieved. Inequality (B.3) is a consequence of the concavity of $b$, the Jensen's inequality, and the definition of conditional mutual information.

(b) This part follows similarly by noting that for any event $E$ we have $\mathbb{E}_Z[\mathrm{Pr}^Z\{E\}] = \mathrm{Pr}\{E\}$. $\square$

Figure 4: PGM representation of processing technique.

## C  Proof of Theorem 4

**Theorem 4** ([9]). *If the random variables $X, Y$ and $Z$ form a Markov chain (in any order), then*

$$I(X; Y|Z) \leq I(X; Y). \tag{C.1}$$

*Proof.* Suppose the conditional dependences of $X, Y$ and $Z$ satisfies a Markov chain $V_1 - V_2 - V_3$ where $V_i \in \{X, Y, Z\}$ and $(V_i)_{i=1}^3$ is a specific ordering of $(X, Y, Z)$. The Markov property results in $V_1 \perp\!\!\!\perp V_3|V_2$. If $V_2 = Z$, $I(X; Y|Z) = I(V_1; V_3|V_2) = 0$ has the smallest possible value and thus (C.1) is satisfied. If $V_2 \neq Z$ either $V_2 = X$ or $V_2 = Y$. Because of the symmetry between $X$ and $Y$, it is enough to consider either of these cases and the other follows similarly. Suppose $V_2 = X$, i.e. $Z \perp\!\!\!\perp Y|X$. By chain rule of mutual information, the quantity $I(Y; (X, Z))$ can be decomposed in two ways and we have

$$I(Y; (X, Z)) = I(Y; Z) + I(Y; X|Z) = I(Y; X) + I(Y; Z|X). \tag{C.2}$$

Since $I(Y; Z|X) = 0$, we have

$$I(X; Y|Z) = I(X; Y) - I(Y; Z). \tag{C.3}$$

Thus, (C.1) is obtained according to the nonnegativity of $I(Y; Z)$. $\qquad\square$

## D  Proof of Lemma 5

**Lemma 5.** *(Processing) Consider random variables $X, Y, \bar{Y}, G, \bar{G}, V, \bar{V}$ and $T$ with the conditional independences presented in the Bayesian network of Fig. 4. Also suppose that $P_{\bar{Y}} = P_Y$, $P_V^Y = P_{\bar{V}}^{\bar{Y}}$ and $P_{\bar{G}}^{T, \bar{V}} = P_G^{T, V}$.*

*(a) If the marginal distribution $P_{\bar{G}}$ is $\sigma$-subgaussian, then*

$$\left| \mathbb{E}[\bar{G} - G] \right| \leq \sqrt{2\sigma^2 I(T; V)}. \tag{D.1}$$

*(b) If $P_{\bar{G}}^t$ is $\sigma$-subgaussian for all $t \in \mathcal{T}$, we have*

$$\Pr\left\{ u \times (\bar{G} - G) \geq \epsilon \right\} \leq \frac{4\sigma^2 (I(T; V) + 1)}{\epsilon^2} \; ; \; u \in \{-1, 1\}. \tag{D.2}$$

*Proof.* Note that $\bar{V} \perp\!\!\!\perp (T, V)$ and marginal distributions $P_V$ and $P_{\bar{V}}$ are the same, because $P_Y = P_{\bar{Y}}$ and $P_V^Y = P_{\bar{V}}^{\bar{Y}}$. The result easilly is obtained as the conditions of Lemma 2 are satisfied for random variables $T, V, \bar{V}, G$ and $\bar{G}$. $\qquad\square$

## E  A Unified View on Information-Theoretic Generalization Bounds

In this section results provided in Section 4.1 are proved and discussed. Recall that we assumed for all $w$ the loss function $\ell(w, X)$ is $\sigma_\ell$-subgaussian. For binary classification with zero-one loss $\sigma_\ell = 1/2$. We also defined $L_j = \ell(W, S_j)$, $L_j' = \ell(W, S_j')$, $R = 1/n \sum_{j=1}^n L_j$ and $R' = 1/n \sum_{j=1}^n L_j'$. Note that $R'$ is $\sigma_\ell^2/n$-subgaussian since $L_j'$s are independent. The object of interest is generalization

Figure 5: Basic generalization.

gap of hypothesis $w$, which is the difference between the expected loss and the average loss on training set

$$\text{gen}(w) = \mathbb{E}_{XY}[\ell(w, (X, Y))] - \frac{1}{n}\sum_{j=1}^{n} \ell(w, S_j). \tag{E.1}$$

We want to have bounds on $\text{gen}(\mu, P_W^S) \triangleq \mathbb{E}[\text{gen}(w)]$ and $\Pr\{\text{gen}(w) \geq \epsilon\}; \epsilon \in \mathbb{R}_+$.

### E.1 Basic Generalization Bound

Considering the conditional independences presented in Fig. 5, using Lemma 2 directly results in

$$\text{gen}(\mu, P_W^S) = |\mathbb{E}[R' - R]| \leq \sqrt{\frac{2\sigma_\ell^2}{n} I(S; W)}, \tag{E.2}$$

and

$$\Pr\{R' - R \geq \epsilon\} \leq \frac{4\sigma_\ell^2(I(S; W) + 1)}{n\epsilon^2}. \tag{E.3}$$

In Section 4.1, we saw that $E[R' - R]$ in the original formulation is equal to $\text{gen}(\mu, P_W^S)$. But the situation is different for $\Pr\{\text{gen}(W) \geq \epsilon\}$ because now the variance also matters. Next theorem proves that for the large enough $N$ this difference just introduces a constant factor to the bound.

**Theorem E.1.** *If $N \geq \frac{2}{\epsilon^2}$ we have*

$$\Pr\{gen(W) \geq \epsilon\} \leq 2\Pr\{R' - R \geq \epsilon/2\}. \tag{E.4}$$

*Proof.* The result follows by a standard technique in proving generalization bound that introduces an auxiliary ghost sample set (for example see Appendix A.1 of [1]). Define $e_\mu(W) = \mathbb{E}_{XY}[\ell(W, (X, Y))]$. If $\Pr\{e_\mu(W) - R \geq \epsilon\} = 0$, the desired inequality holds. Assume $\Pr\{e_\mu(W) - R \geq \epsilon\} > 0$, we have

$$
\begin{aligned}
\Pr\{R' - R \geq \frac{\epsilon}{2}\} &\geq \Pr\{R' - R \geq \epsilon/2 \text{ and } e_\mu(W) - R \geq \epsilon\} \\
&= \Pr\{R' - R \geq \frac{\epsilon}{2} \mid e_\mu(W) - R \geq \epsilon\} \Pr\{e_\mu(W) - R \geq \epsilon\} \\
&\geq \Pr\{e_\mu(W) - R' \leq \frac{\epsilon}{2} \mid e_\mu(W) - R \geq \epsilon\} \Pr\{e_\mu(W) - R \geq \epsilon\} \\
&\geq (1 - e^{-\frac{1}{2}\epsilon^2 n}) \Pr\{e_\mu(W) - R \geq \epsilon\}. \tag{E.5}
\end{aligned}
$$

First inequality holds because taking intersection with another event, does not increase probability. The second inequality is obtained since when $e_\mu(W) - R \geq \epsilon$ and $e_\mu(W) - R' \leq \frac{\epsilon}{2}$ it is guaranteed that $R' - R \geq \frac{\epsilon}{2}$. In the final inequality, Hoeffding's bound is used, which is possible because $W$ is independent of $S'$. □

Using Theorem E.1 we have

$$\Pr\{\text{gen}(W) \geq \epsilon\} \leq \frac{32\sigma_\ell^2(I(S; W) + 1)}{n\epsilon^2}.$$

Figure 6: Conditioning on sample index.

Figure 7: Conditioning on the super sample.

## E.2 Conditioning on Sample Index

Considering Fig. 6, by conditioning on $J$, and processing $S \to S_J$ we have

$$
\begin{aligned}
\text{gen}(\mu, P_W^S) &= \mathbb{E}[L' - L] \\
&\leq |\mathbb{E}[L' - L]| \\
&\leq \mathbb{E}_J[\sqrt{2\sigma_\ell^2 I(S_J; W)}] \\
&= \frac{1}{n}\sum_{j=1}^{n}\sqrt{2\sigma_\ell^2 I(S_j; W)} \qquad\qquad\text{(E.6)} \\
&\leq \sqrt{2\sigma_\ell^2 I(S_J; W)}. \qquad\qquad\text{(E.7)}
\end{aligned}
$$

Here it is assumed that $W$ is independent of the order of samples in $S$ and thus $I(S_J; W|J) = I(S_J; W)$. In Section 4.1, we saw that $\mathbb{E}[L'-L]$ is one of the equivalent ways to compute $\text{gen}(\mu, P_W^S)$, and no further corrections is needed.

## E.3 Conditioning on the Super Sample

Considering Fig. 7, by conditioning on $S^{(1)}, S^{(2)}$ we have

$$
|\mathbb{E}[\bar{R} - R]| \leq \sqrt{\frac{2\sigma_\ell^2}{n} I(U; W|S^{(1)}, S^{(2)})}, \qquad\qquad\text{(E.8)}
$$

and

$$
\Pr\left\{\bar{R} - R \geq \epsilon\right\} \leq \frac{4\sigma_\ell^2(I(U; W|S^{(1)}, S^{(2)}) + 1)}{n\epsilon^2}. \qquad\qquad\text{(E.9)}
$$

Note that here we had to use another branch for random variables $\bar{U}$, $\bar{S}$ and $\bar{R}$ to have the required conditional independence. Moreover, we assumed that the input distribution $\mu$ is continuous, thus $S^{(1)} \cup S^{(2)}$ is a set of $2n$ distinct random variables. As a result, when $S^{(1)} \cup S^{(2)}$ is given, knowing $S$ is the same as knowing $U$ and $I(S; W|S^{(1)}, S^{(2)}) = I(U; W|S^{(1)}, S^{(2)})$. Also note that the discussion on Section 4.1 demonstrated that the distribution of $R$ and $R'$ is the same as in Section E.1. Next theorem shows the relation between the obtained bounds and generalization error.

**Theorem E.2.** *In the setting represented in Fig. 7, we have*

$$
gen(\mu, P_W^S) = 2\,\mathbb{E}[\bar{R} - R]. \qquad\qquad\text{(E.10)}
$$

*Moreover, if zero-one loss is used and $N \geq \frac{64}{\epsilon^2}$, we have*

$$\Pr\{gen(W) \geq \epsilon\} \leq 4 \times \frac{I(U; W|S^{(1)}, S^{(2)}) + 1}{n(\epsilon/8)^2} \tag{E.11}$$

*Proof.* Define $e_\mu(W) = \mathbb{E}_{XY}[\ell(W, (X, Y))]$ and $e_{\hat{\mu}(W)} = \frac{1}{2n} \sum_{(x_i, y_i) \in S^{(1)}, S^{(2)}} \ell(W, (x_i, y_i))$. Note that $e_{\hat{\mu}(W)} = \frac{1}{2}(R + R')$. For expectation of generalization gap, we have

$$
\begin{aligned}
\text{gen}(\mu, P_W^S) &= \mathbb{E}[R' - R] \\
&= \mathbb{E}[2(e_{\hat{\mu}(W)} - R) \\
&= 2\,\mathbb{E}[\bar{R} - R],
\end{aligned} \tag{E.12}
$$

where final equality is due to the fact that $\bar{R}$ is an unbiased estimator for $e_{\hat{\mu}(W)}$, calculated using $n$ samples from $\hat{\mu}$.

For the tail bound we have

$$
\begin{aligned}
\Pr\{\text{gen}(W) \geq \epsilon\} &\leq 2\Pr\{R' - R \geq \epsilon/2\} \\
&= 2\Pr\{e_{\hat{\mu}(W)} - R \geq \epsilon/4\} \\
&= 2\Pr\{e_{\hat{\mu}(W)} - \bar{R}) + (\bar{R} - R) \geq \epsilon/4\} \\
&\leq 2\Pr\{e_{\hat{\mu}(W)} - \bar{R} \geq \epsilon/8\} + \Pr\{\bar{R} - R \geq \epsilon/8\} \\
&\leq 2\left(2e^{-2n(\frac{\epsilon}{8})^2} + \Pr\{\bar{R} - R \geq \epsilon/8\}\right)
\end{aligned} \tag{E.13}
$$

First inequality is based on Theorem E.1. The next equality is based on the algebraic relation $e_{\hat{\mu}(W)} = \frac{1}{2}(R + R')$. In second inequality we used union bound and the mathematical relation $A + B \geq c \Rightarrow (A \geq \frac{c}{2}) \vee (B \geq \frac{c}{2})$. In third inequality, Hoeffding's bound for sampling without replacement is used. Now note that for zero-one loss $\sigma_\ell = 1/2$ and by using (E.9), we have

$$\Pr\left\{\bar{R} - R \geq \epsilon/8\right\} \leq \frac{I(U; W|S^{(1)}, S^{(2)}) + 1}{n(\epsilon/8)^2}. \tag{E.14}$$

On the other hand, when $N \geq \frac{64}{\epsilon^2}$ we also have

$$2e^{-2n(\frac{\epsilon}{8})^2} \leq \frac{1}{n(\epsilon/8)^2}. \tag{E.15}$$

This inequality can be verified for $N = \frac{64}{\epsilon^2}$. Which means that it is also valid for larger $N$, because l.h.s approaches zero with exponential rate while the r.h.s has slower rate of $\frac{1}{n}$.

Finally, we have

$$2e^{-2n(\frac{\epsilon}{8})^2} + \Pr\{\bar{R} - R \geq \epsilon/8\} \leq 2 \times \frac{I(U; W|S^{(1)}, S^{(2)}) + 1}{n(\epsilon/8)^2}, \tag{E.16}$$

which concludes the proof. □

### E.4 Conditioning on the Super Sample and Index

Considering Fig. 6, by conditioning on $J, S^{(1)}, S^{(2)}$, and processing $S \to S_J$ we have

$$
\begin{aligned}
|\mathbb{E}[\bar{L} - L]| &\leq \mathbb{E}_J[\sqrt{2\sigma_\ell^2 I^J(U_J; W|S^{(1)}, S^{(2)})}] \tag{E.17} \\
&\leq \sqrt{2\sigma_\ell^2 I(U_J; W|S^{(1)}, S^{(2)}, J)} \tag{E.18}
\end{aligned}
$$

Note that the same argument which was used in Theorem E.2, can be used for datasets with one sample to show that $\text{gen}(\mu, P_W^S) = 2\,\mathbb{E}[\bar{L} - L]$.

Figure 8: Conditioning on the super sample and index.

# F Proof of Theorem 6

**Definition 1** (Separable process). *The random process $\{X_t\}_{t \in T}$ is called separable if there is a countable set $T_0 \subseteq T$ such that $X_t \in \lim_{\substack{s \to t \\ s \in T_0}} X_s$ for all $t \in T$ a.s., where $x \in \lim_{\substack{s \to t \\ s \in T_0}} x_s$ means that there is a sequence $(s_n)$ in $T_0$ such that $s_n \to t$ and $x_{s_n} \to x$.*

**Theorem 6.** *Assume that $X_{\mathcal{W}} = \{\sqrt{n}\mathrm{gen}(w)\}_{w \in \mathcal{W}}$ is a separable subgaussian process on the bounded metric space $(\mathcal{W}, d)$ and the learned hypothesis $W$ is a deterministic function of $X_{\mathcal{W}}$. Consider the sequence of functions $(\Pi_k)_{k=k_1(\mathcal{W})}^{\infty}$ where $k_1(\mathcal{W})$ is the largest integer that satisfies $2^{-(k_1(\mathcal{W})-1)} \geq diam(\mathcal{W})$, and for all $k \geq k_1$, $\Pi_k : \mathcal{W} \to \mathcal{W}$ is a function satisfying $d(w, \Pi_k(w)) \leq 2^{-k}; \forall w \in \mathcal{W}$. Define $\tilde{W}_k = \Pi_k(W)$, we have*

$$gen(\mu, P_W^S) \leq \frac{1}{\sqrt{n}} 6\sqrt{2} \sum_{k=k_1(\mathcal{W})}^{\infty} 2^{-k} \sqrt{I(\tilde{W}_k; S)}. \tag{F.1}$$

*Proof.* This result is based on Theorem 11 of [3]. Here it is restated using notations used in current paper.

**Theorem F.1.** *Assume that $X_{\mathcal{W}} = \{X_w\}_{w \in \mathcal{W}}$ is a separable subgaussian process on the bounded metric space $(\mathcal{W}, d)$. Consider the sequence of functions $(\Pi_k)_{k=k_1(\mathcal{W})}^{\infty}$ where $k_1(\mathcal{W})$ is the largest integer that satisfies $2^{-(k_1(\mathcal{W})-1)} \geq diam(\mathcal{W})$, and for all $k > k_1$, $\Pi_k : \mathcal{W} \to \mathcal{W}$ is a function satisfying $d(w, \Pi_k(w)) \leq 2^{-k}; \forall w \in \mathcal{W}$. Define $\tilde{W}_k = \Pi_k(W)$ for $k \geq k_1$ and $\tilde{W}_{k_1-1} = w_0$ for an arbitrary $w_0 \in \mathcal{W}$. We have*

$$\mathbb{E}[X_W] \leq 3\sqrt{2} \sum_{k=k_1(\mathcal{W})}^{\infty} 2^{-k} \sqrt{I(\tilde{W}_{k-1}, \tilde{W}_k; X_{\mathcal{W}})}. \tag{F.2}$$

Theorem F.1 is stated for a general random process. In Theorem 6, we used $X_W = \{\sqrt{n}\mathrm{gen}(w)\}_{w \in \mathcal{W}}$. Moreover, knowing $S$, the values $\{\sqrt{n}\mathrm{gen}(w)\}_{w \in \mathcal{W}}$ are all deterministically calculated. Thus, by data processing inequality $I(\tilde{W}_{k-1}, \tilde{W}_k; X_{\mathcal{W}}) \leq I(\tilde{W}_{k-1}, \tilde{W}_k; S)$. Now the only remained part is

to remove the dependence in $\tilde{W}_{k-1}$ without resorting to the nested partitioning. We have

$$
\begin{aligned}
\mathbb{E}[X_W] &\leq 3\sqrt{2}\sum_{k=k_1}^{\infty} 2^{-k}\sqrt{I(\tilde{W}_{k-1}, \tilde{W}_k; X_{\mathcal{W}})} \\
&= 3\sqrt{2}\sum_{k=k_1}^{\infty} 2^{-k}\sqrt{I(\tilde{W}_k; X_{\mathcal{W}}) + I(\tilde{W}_{k-1}; X_{\mathcal{W}}|\tilde{W}_k)} \\
&\leq 3\sqrt{2}\sum_{k=k_1}^{\infty} 2^{-k}\sqrt{I(\tilde{W}_k; X_{\mathcal{W}}) + I(\tilde{W}_{k-1}; X_{\mathcal{W}})} && \text{(F.3)} \\
&\leq 3\sqrt{2}\Big(\sum_{k=k_1}^{\infty} 2^{-k}\sqrt{I(\tilde{W}_k; X_{\mathcal{W}})} + \sum_{k=k_1}^{\infty} 2^{-k}\sqrt{I(\tilde{W}_{k-1}; X_{\mathcal{W}})}\Big) && \text{(F.4)} \\
&\leq 3\sqrt{2}\Big(\sum_{k=k_1}^{\infty} 2^{-k}\sqrt{I(\tilde{W}_k; X_{\mathcal{W}})} + \sum_{k=k_1-1}^{\infty} 2^{-(k+1)}\sqrt{I(\tilde{W}_k; X_{\mathcal{W}})}\Big) \\
&\leq 3\sqrt{2}\Big(\sum_{k=k_1}^{\infty} 2^{-k}\sqrt{I(\tilde{W}_k; X_{\mathcal{W}})} + \sum_{k=k_1-1}^{\infty} 2^{-k}\sqrt{I(\tilde{W}_k; X_{\mathcal{W}})}\Big) \\
&\leq 6\sqrt{2}\sum_{k=k_1}^{\infty} 2^{-k}\sqrt{I(\tilde{W}_k; X_{\mathcal{W}})}. && \text{(F.5)}
\end{aligned}
$$

Inequality (F.3) is valid based on Theorem 4 because $\tilde{W}_{k-1} \perp\!\!\!\perp \tilde{W}_k | X_{\mathcal{W}}$ (which is a consequence of the deterministic relation between $W$ and $X_{\mathcal{W}}$). In (F.4) we used the mathematical relation $\sqrt{a+b} \leq \sqrt{a} + \sqrt{b}; \forall a, b \in \mathbb{R}_+$. In final inequality we used $I(w_0; X_{\mathcal{W}}) = 0$, because $w_0$ is a deterministic value. $\qquad\square$

## G  Proof of theorem 7

**Theorem 7.** *Suppose $\mathcal{W}$ has VC-dimension $d_{(vc)}(\mathcal{W})$ and $\ell(w, (x,y)) = \mathbb{1}(w(x) \neq y)$. There is a universal constant $C$ such that the following bound holds on $R_U(D)$ defined in (28)*

$$
R_U(D) \leq C d_{(vc)}(\mathcal{W}) \log(\frac{C}{D}). \tag{G.1}
$$

*Proof.* Note that $I^{S^{(1)}S^{(2)}}(\tilde{W}_k, S) \leq H(\tilde{W}_k)$, where $H(\tilde{W}_k)$ is the entropy of random variable $\tilde{W}_k$ and is bounded by logarithm of number of possible values it takes. A possible mapping for producing $\tilde{W}$ is to use centers of a $D$-covering. Thus,

$$
R^{S^{(1)}S^{(2)}}(D) \leq \log N(\mathcal{W}, d^{S^{(1)}S^{(2)}}, D), \tag{G.2}
$$

where $N(\mathcal{W}, d^{S^{(1)}S^{(2)}}, D)$ is the covering number of $(\mathcal{W}, d)$ at the scale $D$. Recall that $d^{S^{(1)}S^{(2)}}(w, w') = 2\|w - w'\|_{L^2(\hat{\mu})}$. Here $\hat{\mu}$ is the empirical distribution of the superset and

$$
\|w - w'\|_{L^2(\hat{\mu})} = \left[\frac{1}{n}\sum_{j=1}^{n}(w(x_j) - w'(x_j))^2\right]^{1/2}. \tag{G.3}
$$

Now we use the following theorem which is based on [10].

**Theorem G.1** (Dudley [10])**.** *There is a universal constant $C$ such that*

$$
\sup_{\mu} N(\mathcal{W}, \|.\|_{L^2(\mu)}, D) \leq \left(\frac{C}{D}\right)^{Cd_{(vc)}(\mathcal{W})} ; \quad \forall D < 1.
$$

Using this theorem with inequality (G.2) completes the proof. $\qquad\square$