[Reviews · NeurIPS 2020]

Review 1

Summary and Contributions: This paper presents a probabilistic graphical representation approach to obtaining information-theoretic generalization bounds and proposes two general techniques of "conditioning" and "processing." Some recent results in the literature are explained with this proposed technique. The paper also proposes and studies the "chaining conditional mutual information" bound and shows an advantage of that bound over the recent conditional mutual information bound.

Strengths: - The paper studies information-theoretic generalization bounds and presents new techniques. The topic of generalization bounds in statistical learning is of major interest to the NeurIPS community. - Using probabilistic graphical representations for interpreting generalization bounds and as a potential framework for obtaining new bounds is new and interesting and the techniques of processing and conditioning build upon recent results in the literature. These types of visualizations may lead to new and useful generalization bounds in follow-up works. Also, combining "chaining" with "conditional mutual information" bound is new and expands the venue of combining information-theoretic techniques with probabilistic techniques for obtaining new tools in statistical learning. - The claims in the paper seem to be sound, except for one result (Theorem 6) that I did not completely understand and I mention below.

Weaknesses: 1. In Section 4, conditioning and processing are proposed as a general framework for giving a "unified view" on information-theoretic generalization bounds, however, in Section 5 to illustrate an application, suddenly the method of chaining is brought into picture which is outside that framework. In other words, it seems that the two parts of the paper are fairly disconnected. The only examples given from purely using the processing and conditioning tools correspond to known results in the literature such as [5, 15] as mentioned in the paper. It can make the paper stronger if further examples of new generalization bounds can be provided purely based on the proposed techniques of conditioning and processing. 2. What is stated as Theorem 4 and proved in the appendix - without citing a reference - is actually an old result in information theory (which should not be surprising given its simple form). For example, see Corollary of Theorem 2.8.1 in the book by Cover-Thomas (p. 35). A reference should be cited for this result. 3. The chaining conditinal mutual information bound seems to be interesting, though the only benefit of this new bound that the paper demonstrates, if I understood correctly, is that it can deduce the uniform bound of VC-dimension, while conditional mutual information will be "log n" factor bigger. This is then seems similar to the discussion in Section 28.1 of Shalev-Schwartz & Ben-David book about dropping the log factor by using the method of chaining. However, this does not look like a satisfying application since if we wanted to obtain uniform bounds such as VC-dimension then why should we use mutual information to begin with? It would make the paper stronger if any advantage of that bound can be shown over conditional mutual information in some examples, apart from uniform bounding.

Correctness: The results appear correct to me, except for something in the proof of Theorem 6 that I did not completely understand and would like the author(s) to explain in their response: How is inequality (F.3) deduced from Theorem 4? The inequality involves X_{W} while the Markov chain includes W. ----- Post Author Response ----- I decided to keep my score at 6 since I did not find the author response to the mistake in the proof of Theorem 6 (inequality F.3) adequate. The statement of Theorem 6 is likely false, as one should take into account the interaction between the scales of the chaining bound either by assuming hierarchical coverings such as Theorem 3 of https://arxiv.org/pdf/1806.03803.pdf or showing such interaction explicitly in the bound as in Theorem 11 of the aforementioned paper, a result which this paper can cite and use instead of Theorem 6. Assuming a deterministic algorithm makes the result too weak and restricted; in fact there is no need to use mutual information I(W_k;S) in this case as they will all equal the entropy of the marginal of the algorithm output H(W_k).

Clarity: The paper is well written except for some typos and minor comments (see below).

Relation to Prior Work: The paper has mentioned previous contributions, though it needs to be improved by elaborating more on the relation between its contributions and prior work, specifically the discussion in the paragraph 159-169. For example, in line 163 it is written that "it is not exactly what we looked for" which needs further explanation. Similarly in line 168 about the difference in constant factors (which I did not find in the appendix) and in line 179 about the difference on the learning algorithm assumptions. Also, in Section 5, the author(s) should compare the chaining conditional mutual information bound with the work by Audibert-Bousquet '07 ("Combining PAC-Bayesian and Generic Chaining Bounds") which also uses double sample sets and chaining.

Reproducibility: Yes

Additional Feedback: Minor comments: - Notations --> Notation - Let's --> Let us (please avoid contractions) - 68: Is there a typo in the equation having P^{XY}_G? - Is the Slepian-Wolf work [14] used anywhere in the paper? - 331: chain rule of mutual information.


Review 2

Summary and Contributions: 1. This paper provides a unifying framework to describe recent work on information-theoretic generalization bounds. In particular, it unifies conditional/disintegrated mutual information bounds, chaining mutual information and leave-one-out style bounds using two key steps of conditioning and preprocessing, and repeated nested applications of these procedures. 2. The paper (partially) resolves (a variant of) the conjecture of Steinke and Zakynthinou of whether information theoretic generalization bounds based on their CMI can recover tight generalization bounds based on VC-theory in the non-realizable case.

Strengths: 1. The results appear to be sound. 2. The graphical model representation is a convenient, intuitive and useful representation of the various types of techniques for information theoretic generalization bounds which have emerged in recent years. This unification will have a significant impact in the communities ability to deploy modular combinations of those results to derive tight bounds for various learning problems and algorithms. 3. The positive (partial) resolution of (a version of) the conjecture of Steinke and Zakynthinou is an important contribution which essentially validates that an information theoretic approach to generalization is at least as expressive as an approach based on uniform uniform convergence (a.k.a agnostic PAC/ Uniform Gliovenko-Cantelli). This will likely have a long term impact on the prevailing methods researchers will use for bounding generalization error in future work.

Weaknesses: 1. I did not find any particular weaknesses of the work that need to be addressed.

Correctness: The claims appear to be correct. POST REBUTTAL COMMENTS: Regarding R1's review correctly pointing out the error in (F3), I feel the author response fell a bit short. Rather than assuming that the learning algorithm is deterministic, which would be a problem for the main algorithms of interest (variants of sgd). The authors could have instead used nested coverings, as in Theorem~3 of https://arxiv.org/pdf/1806.03803.pdf, or used something like Theorem~11 of the same work. The result could have just been cited instead of being reproved. The determinism assumption they mentioned in rebuttal is much stronger than necessary and would adversely impact the relevance of the work. Either way, due to the error, and the weak author response, it leads me to lower my score to an 8 instead of a 9. The error must be corrected in the final version.

Clarity: 1. The paper is very easy to read and does not require specialist knowledge to understand.

Relation to Prior Work: 1.The relation to the most related prior work is thorough enough, 2. though the article would benefit from a more broad literature review if space permits in the final version. 3. It would be helpful if the exact relationship between Thm.~6,7 and the conjecture of Steinke and Zakynthinou (both from their original paper, and from the more recent preprint about various forms of the conjecture) was more clearly and explicitly expressed. This work resolves their conjecture in spirit, but not according to its precise statement, as far as I can tell, because it does not bound the CMI of their learning algorithm directly, but instead bound other information theoretic quantities that are shown to be more immediately relevant.

Reproducibility: Yes

Additional Feedback: 1. It would be helpful Table 1 included the relevant papers as a fourth column. Can Negrea et al. [10] be included in the table as well? 2. Typo "=\le" in ldisplay equation above line 171 3. In remark 1, it says that (b) is stronger than (a). But (b) does not necessarily imply (a) since if $X$ is cauchy and $P^x_{\bar G}$ is $N(x,1)$ then $P_{\bar G}$ is not subG. 4. The impact statement is underwhelming. This can easily be improved. 5. My review score (9) is based on the current state of the manuscript and my interpretation of it. Since my only feedback/suggestions are very minor and my current impression of the work is already very high, I don't foresee myself increasing my score unless the authors demonstrate in the rebuttal that I've missed or underestimated an important contribution of the work. POST REBUTTAL COMMENTS: I have reduced my score to an 8 instead of a 9, based on the error pointed out by R1. See my comments in the "Correctness" field for further details.


Review 3

Summary and Contributions: This paper introduces two techniques that can improve existing information-theoretical generalization bound. The probabilistic graphs shown in this work illustrate how the techniques can be applied. Other results include an extension to chaining mutual information and recovering the VC bound via information-theoretical tools.

Strengths: This paper deals with the high-level ideas of how to improve existing information-theoretical generalization bound. The conditioning and processing techniques could be novel to some in the machine learning community and potentially lead to concrete improvement on existing bounds.

Weaknesses: Most of the contents in this paper could be too high level to claim any concrete contributions. While it is claimed the techniques can help improving bounds, it is not clear how exactly it would improve existing bounds. The conditioning technique can provide an unconditioned bound, given a conditioned variant. It is not very clear how much the mutual information will decrease. In other words, this paper could be very inspiring, but not quite ready to appear as a concrete conference paper. More and stronger theoretical results that show improvement are needed.

Correctness: The derivations are correct.

Clarity: While the presentation is easy to follow, it is a bit unclear to identify the important results and its contribution.

Relation to Prior Work: Yes.

Reproducibility: Yes

Additional Feedback: Post-rebuttal: I have read the response. Given the authors' response I will raise my score to 6. ================================== I find it hard to imagine how the bound is improved via conditioning and processing. Is there any concrete examples that shows the bound is indeed improved?


Review 4

Summary and Contributions: This paper studied the improvement of information-theoretic generalization bounds. Additional on traditional information-theoretic generalization bounds which only considered a pair of random variables X and Y, this work aims to improve the bound to more random variables. Two techniques --- conditioning and processing --- are introduced. Conditioning upper bound is given by conditional MI I(X:Y|Z) for conditioning random variable Z. Processing upper bound is given by I(T;V) where T and V are processed random variables of X and Y respectively. These techniques can be used to provide a general framework for understanding information-theoretic generalization bounds.

Strengths: Information-theoretic generalization bounds have been actively studied recently. The idea of improving these bounds to graphical models are novel and important. This paper is the first step toward a general improvement --- information-theoretical generalization bound for complex graphical model or complex neural network. This is relevant to NeurIPS community.

Weaknesses: The mathematical content of this paper is not very novel. It is highly dependent on traditional information-theoretic generalization bound by Xu and Raginsky, plus information-processing inequality and conditioning decreases mutual information, which are standard results in information theory. It will require more work if we want to extend the information-theoretical generalization bounds to more complex graphical models

Correctness: I don't find any error in the proofs in this paper.

Clarity: Yes, this paper is written clearly.

Relation to Prior Work: This paper have cite most of the recent works about information-theoretic generalization bounds. It is better if the authors could introduce some of the recent application of information-theoretic generalization bounds to emphasize the practical usage of these bounds and the importance of the improvement in this paper. Recommended reference: Information-Theoretic Understanding of Population Risk Improvement with Model Compression. Y Bu, W Gao, S Zou, VV Veeravalli Generalization bounds of sgld for non-convex learning: Two theoretical viewpoints W Mou, L Wang, X Zhai Understanding autoencoders with information theoretic concepts S Yu, JC Principe

Reproducibility: Yes

Additional Feedback:

[Author Response · NeurIPS 2020]

We thank the reviewers for the time they have spent on this work and their valuable comments. We will try our best to make the suggested improvements in the final version. Below please find the responses to the main raised comments.

•**R1: Section 5 seems disconnected.** The results of Section 5 are based on a direct application of "conditioning" (Lemma 3), and thus a necessary element of our work. Note that Lemma 3 is a quite general result which is applicable to any concave information-theoretic bound. Its application along with Lemma 2 produces the results of Section 4. But it is not restricted to this case (the title of Section 4 will be changed to eliminate this confusion). To demonstrate this, we decided to apply the technique on the chaining bound of [2] which might seem quite different. Note that the bound of Eq. (21) is a concave function of $I(\tilde{W}_k; S), \forall k$ and this is enough to apply conditioning (as was done in Eq. 26). Another application of conditioning, omitted due to space constraints, is a generalization of Lemma 1 which uses Cumulant Generating Functions (CGF) and the square root function is replaced by another concave function (the inverse of Legendre dual of CGF in case it exists) and conditioning technique still applies.

•**R1: Inequality F.3.** Thank you for this careful observation. This inequality is correct under the assumption that $W$ is a deterministic function of $X_{\mathcal{W}}$ (e.g. a deterministic ERM); i.e., we have $\tilde{W}_{k-1} \perp\!\!\!\perp \tilde{W}_k | W \Rightarrow \tilde{W}_{k-1} \perp\!\!\!\perp \tilde{W}_k | X_{\mathcal{W}}$. This assumption will be explicitly stated in the theorem and the text will be updated adequately. The rest of the reasoning of Section 5 for proving the generalization bound remains unchanged (actually the term "rate-distortion" was used taking the deterministic setting in mind, as stated in Line 199). Note that for non-deterministic algorithms, $I(\tilde{W}_{k-1}, \tilde{W}_k; X_{\mathcal{W}})$ in Eq. (F.2) can be bounded by $I(\tilde{W}_{k-1}, \tilde{W}_k; W)$ which yields similar results for $I(\tilde{W}_k; W)$ instead of $I(\tilde{W}_k; S)$.

•**R1: Theorem 4.** This theorem is a regathering of simple information-theoretic inequalities (such as the mentioned corollary of Cover) to summarize some conditions on the graphical model which can result in conditional mutual information being less than its unconditional counterpart. We will emphasize on this and add the related reference.

•**R1: Beyond uniform bounds.** The main purpose of Section 5 is to show that by utilizing the conditioning technique it is possible to close the gap between the classic generalization bounds of VC-theory and information-theoretic bounds. Though the final results of that section do not provide new bounds, it shows that the information-theoretic approach is at least as strong as the VC-theory (a question which was frequently raised in this line of work and partially addressed in [15]). It is worth mentioning that we are working on more advanced usages of these techniques for future work.

•**R1: Literature review.** We will elaborate more about the mentined paragraphs. We will also include discussions about Audibert and Bousquet papers and some discussions about PAC-Bayesian bounds in general.

•**R1 & R2: Relation with Steinke and Zakynthinou [15].** We think an approach similar to the one discussed in Section 5 can be used to answer the first conjecture presented by that paper (as this is another track we are following, we should add that the connection is not trivial). But, here we want to clarify the differences in our problem settings. Their work on VC-dimension and the conjectures therein are about proving the "existence of an ERM algorithm" which has a small CMI. On the other hand, they also show that CMI can actually be quite large for some other ERM algorithms and it is not controlled by $d_{vc}$ in general (not even a bound with $\log(n)$ factor exists). Thus, there are some ERM algorithms whose generalization cannot be explained by CMI, which shows that fundamentally CMI is not as expressive as the standard uniform convergence. In their first conjecture (the one related to our work in Section 5), they even had to go beyond ERM algorithms and allow $\epsilon$ empirical risk. Our work does not have these limitations. To do that, we had to go beyond regular CMI and also incorporate processing (Section 4.2) and chaining (Section 5) techniques.

•**R2: Extending table.** As suggested, we will try to include the results of Negrea et al. [10] in Table 1 (it is an extension of the second row which in this case a group of indices are conditioned) and also a column for references.

•**R3: Demonstrating that bounds are tightened.** We acknowledge that the discussion on this matter is better to be clarified in the final version. But, these techniques can indeed tighten the bounds: 1) in Section 4 it is demonstrated that the results of [5] and [8] on tightening the mutual information bounds can be explained using the proposed framework in a straightforward manner. There are some theoretical and empirical discussions on these papers demonstrating that these bounds can be (much) tighter. These were not included in the text due to space constraints. We will add some elaboration on this. 2) basic mutual information bounds can be very large even for simple hypothesis sets (as demonstrated in [4]). This problem was partially addressed in [15] and was followed in the current work by proving optimal bounds comparable to VC-theory (both of these approaches use the conditioning Lemma at their core). This is another application of using these techniques to improve the bounds.

•**R3 & R4: Concrete and recent practical applications.** We can use the extra page to discuss some recent applications of the bounds in practice (in particular more discussion on SGLD which was suggested by reviewers 4). We believe this discussion will demonstrate how such tightened bounds can be used in practice to analyze neural networks.

•**R4: Mathematical novelty.** It is hard to claim true mathematical "novelty", but here we summarize some of our contributions: 1) Extending Lemma 1 to stochastic mappings and also providing general formulations for tail bounds, 2) formulation of conditioning Lemma as a general technique explaining a variety of previous results and applicable to new situations, 3) characterization of the chaining mutual information bound as a (variant of) rate-distortion problem, 4) combining chaining and conditioning to close the gap between information-theoretic and uniform convergence bounds.

•**R4: More complex graphical models.** The studied graphical models already have a variety of applications, as discussed in Sections 4 and 5. But, we agree that much is remained to be explored in future work.

[Meta-Review · NeurIPS 2020]

The paper discusses generalization bounds based on information-theoretic notions. The techniques developed here may offer some advantages over the recently-proposed conditional mutual information (CMI) approach; in particular, they compose the CMI technique with the "chaining" technique to recover the known sharp generalization bounds in agnostic learning with VC classes, whereas it is still open whether such sharp bounds are achievable via CMI directly. The reviewers all favor acceptance. A reviewer did find a mistake in the paper, which the authors acknowledge in their response, and propose a fix; however, the reviewers have other suggestions for better corrections in their post-rebuttal updates. The reviews also note that a couple of the results are known or follow from known results (though others are new and interesting). Reviewers also noted a lack of concrete examples where these techniques offer provable advantages over all existing analyses.